# Achieving Sample and Computational Efficient Reinforcement Learning by Action Space Reduction via Grouping

**Yining Li** [*], **Peizhong Ju**[*],**& Ness Shroff**[*,†]
[*]Department of Electrical and Computer Engineering
[†]Department of Computer Science and Engineering
The Ohio State University
Columbus, OH 43210, USA
{li.12312, ju.171, shroff.11}@osu.edu

## Abstract

Reinforcement learning often needs to deal with the exponential growth of states and actions when exploring optimal control in high-dimensional spaces (often known as the curse of dimensionality). In this work, we address this issue by learning the inherent structure of action-wise similar MDP to appropriately balance the performance degradation versus sample/computational complexity. In particular, we partition the action spaces into multiple groups based on the similarity in transition distribution and reward function, and build a linear decomposition model to capture the difference between the intra-group transition kernel and the intra-group rewards. Both our theoretical analysis and experiments reveal a *surprising and counter-intuitive result*: while a more refined grouping strategy can reduce the approximation error caused by treating actions in the same group as identical, it also leads to increased estimation error when the size of samples or the computation resources is limited. This finding highlights the grouping strategy as a new degree of freedom that can be optimized to minimize the overall performance loss. To address this issue, we formulate a general optimization problem for determining the optimal grouping strategy, which strikes a balance between performance loss and sample/computational complexity. We further propose a computationally efficient method for selecting a nearly-optimal grouping strategy, which maintains its computational complexity independent of the size of the action space.

## 1 Introduction

Reinforcement learning (RL), a field dedicated to finding the optimal policy that maximizes the long-term return through interactions with the environment, suffers from "the curse of dimensionality" (Barto and Mahadevan, 2003). In other words, in high-dimensional scenarios, the state-action space of RL grows exponentially with the number of degrees of freedom. For instance, in a control system, there could be millions of potential actions available at each step. Similarly, within a large system, a recommender system might have to consider millions of items (Dulac-Arnold et al., 2015). This exponential growth poses a significant complexity barrier to discovering optimal policies, especially in large-scale systems (Azar et al., 2012; Agarwal et al., 2020b).

To overcome the challenges associated with the explosion of the state-action space, a common approach is to use a low-rank representation of the Markov Decision Process (MDP). In low-rank MDP settings that allow for polynomial sample complexity relative to the horizon length and feature dimension, some works investigate simultaneous learning of representations and the optimal policy (Agarwal et al., 2020a; Modi et al., 2020). However, the existing literature often assumes the attainability of an exact low-rank representation of the state-action space, wherein the representation accurately reflects the MDP's characteristics. In practical scenarios, low-rank structures are often corrupted by noise, but the literature does not consider the errors resulting from the mismatch between the low-rank representation and the MDP itself.

In order to address the aforementioned limitations, researchers have explored the use of abstractions, which involves learning the low-dimensional latent state/action space that throws away some irrelevant state/action features. Previous literature has investigated various similarity metrics to identify suitable abstractions aligned with the original MDP, such as model similarity (Jiang et al., 2015; Gelada et al., 2019) and the similarity of the optimal value function (Abel et al., 2016; 2020). Some studies have investigated the performance degradation resulting from inaccurate abstractions (Abel et al., 2020). To our knowledge, existing literature does not address the both sample complexity and computational complexity associated with estimating abstractions.

Our work falls within the realm of learning abstractions to preserve the performance of the optimal policy. We specifically focus on MDPs with large action spaces that exhibit group-wise similarity, where only approximate grouping strategies of the action space are available. Leveraging abstractions of the underlying MDP offers the advantage of reduced complexity, albeit at the cost of worse performance due to the approximation. This raises a fundamental question: *How does the trade-off between complexity reduction benefits and performance loss manifest in the context of an MDP and its corresponding grouping strategy?*

Interestingly, our analysis yields a counter-intuitive result: *while a finer grouping strategy minimizes model approximation errors, the utilization of sample-based estimation also contributes to the performance shortfall,* especially when faced with limited samples and computational resources. This further prompts the question of how to select a grouping strategy that strikes a balance between approximation error and estimation error to minimize the performance loss.

Our main contributions are as follows.

- We propose an action-grouping method that clusters actions based on the similarity between intra-group transition kernels and rewards. This grouping approach allows us to effectively reduce the size of the action space. Furthermore, we ensure that the performance degradation remains within acceptable bounds by carefully selecting the grouping strategy.

- We analyze the performance loss, taking into account both approximation errors caused by information loss of grouping and estimation errors caused by limited sampling and computational resources. We compare our result with a known lower bound on the estimation error to show that it is tight. We then provide an example for which the approximation error is also relatively tight. We further give some insights into understanding the relationship between the grouping function and performance loss.

- We build a general optimization problem over the grouping function, sample size, and iteration number, enabling us to achieve a balance between performance degradation and sample/computational complexity. The complexity of finding the optimal grouping is proportional to the number of feasible groups.

## 2 RELATED WORK

To avoid the curse of dimensionality in tabular MDP, there have been several works on learning and exploring the inherent structure of large MDP.

**Representation Learning in Low-rank MDP**  One line of work is to consider reducing sample efficiency by exploring MDP structures. Several studies have explored the sufficient and necessary conditions for learning nearly-optimal policies with polynomial sample efficiency, relative to the horizon length and feature dimension (Jiang et al., 2017; Sun et al., 2019; Du et al., 2021; Weisz et al., 2021). In MDP settings that allow for polynomial sample complexity, such as low Bellman rank (Jiang et al., 2017; Wang et al., 2021; Ayoub et al., 2020; Zhou et al., 2021; Du et al., 2019; Agarwal et al., 2020a), low witness rank (Sun et al., 2019), and bilinear structure (Du et al., 2021), some works assume that the agent possesses knowledge of a low-rank representation and focus on exploration algorithms (Jiang et al., 2017; Wang et al., 2021; Li et al., 2023). A more practical approach is to learn a good latent representation of specially-structured MDPs through rich observations (Du et al., 2019; Agarwal et al., 2020a; Modi et al., 2020; Uehara et al., 2021; Zhang et al., 2022). However, the above-mentioned feature selection algorithms are based on the realizability assumption, which assumes there exists an exact mapping function from the latent state space to observations. In contrast,

our setting does not assume that this exact mapping exists, and we allow optimizing the grouping function which belongs to the given feasible set.

**State/action Abstractions**    Another line of work learns abstractions, which do not hold assumptions on the specific structures of latent state/action space. The abstractions can be categorized into state, action, and joint state/action abstractions. For the state abstractions, Li et al. (2006) build a uniform model of state abstraction to preserve enough information to find good policies. It generalizes different types of state abstractions, such as bisimulation (Givan et al., 2003), and homomorphisms (Ravindran and Barto, 2002). Ravindran and Barto (2004) emphasize the performance loss by the MDP approximation. Li et al. (2006) also provide the convergence performance of the resulting abstract policy in the ground MDP. Abel et al. (2016) further investigate the performance guarantee of approximate state abstractions which treats nearly-identical states as equivalent. Several algorithms have been proposed to select a good state abstraction from a given set of feasible abstractions (Jiang et al., 2015; Ortner et al., 2019).

A well-researched action abstraction type is options which are temporally related actions from an initial state and terminal state (Sutton et al., 1999). A trend of joint state-action abstraction is the hierarchical abstraction design, in which higher-level policies communicate between goals (subspace of state spaces) and lower-level policies aim to achieve goals from initial states (Nachum et al., 2019; Abel et al., 2020; Jothimurugan et al., 2021). Specifically, Abel et al. (2020) learns the performance loss associated by pairing the options with state abstractions and presents the sufficient and necessary conditions for options to preserve information for nearly-optimal policy. There are also works on learning latent state space models end-to-end using neural networks (Hafner et al., 2019; Ha and Schmidhuber, 2018; Gelada et al., 2019).

However, previous literature on abstraction learning primarily focuses on the performance loss resulting from approximate abstractions, while ignoring the complexity including both sample complexity and computational complexity. In our work, we address this gap by considering both the performance loss and the sample/computational complexity when determining the optimal grouping function.

## 3 SYSTEM MODEL

### 3.1 MDP PRELIMINARIES

This paper focuses on infinite-horizon discounted Markov Decision Processes (MDP) $\mathcal{M} := (\mathcal{S}, \mathcal{A}, \mathbb{P}, R, \gamma)$. Both $\mathcal{S}$ and $\mathcal{A}$ are discrete and finite. Here, $\mathcal{S}$ and $\mathcal{A}$ represent the state and action space, with sizes denoted as $S$ and $A$, respectively. $\mathbb{P} : \mathcal{S} \times \mathcal{A} \to \Omega(\mathcal{S})$ is the transition kernel, where $\Omega(\mathcal{S})$ is the collection of probability distributions over state space $\mathcal{S}$. $\mathbb{P}(\boldsymbol{s}'|\boldsymbol{s}, \boldsymbol{a})$ represents the probability of transiting to state $\boldsymbol{s}'$ when the agent plays action $\boldsymbol{a}$ at state $\boldsymbol{s}$. $R(\boldsymbol{s}, \boldsymbol{a})$ is the instant reward with state-action pair being $(\boldsymbol{s}, \boldsymbol{a})$. We have the following assumption on the rewards, which is commonly used in RL (Antos et al., 2008; Wang et al., 2021).

**Assumption 1.** *(Bounded rewards) Assume that the reward satisfies $0 \leq R(\boldsymbol{s}, \boldsymbol{a}) \leq 1$ for any state-action pair $(\boldsymbol{s}, \boldsymbol{a})$.*

The policy on $\mathcal{M}$ is defined as a mapping from $\mathcal{S}$ to the probability distributions over the action space, i.e., $\pi : \mathcal{S} \to \Omega(\mathcal{A})$. Let $Q_{\mathcal{M}}(\boldsymbol{s}, \boldsymbol{a})$ and $V_{\mathcal{M}}(\boldsymbol{s})$ denote the value function based on the policy $\pi$ from the initial state-action pair $(\boldsymbol{s}, \boldsymbol{a})$ and $\boldsymbol{s}$, respectively. There exists an optimal policy $\pi_{\mathcal{M}}^*$ that maximizes the value function simultaneously for each state, and the state-action value function based on policy $\pi_{\mathcal{M}}^*$ is the fixed point of the Bellman optimality operator (Puterman, 2014). For notational simplicity, we write the value function based on the optimal policy as $Q_{\mathcal{M}}^*$ and $V_{\mathcal{M}}^*$ in the following.

### 3.2 ACTION GROUPING

To capture the similarity characteristics, we assume actions can be classified into multiple groups based on prior knowledge of $\mathcal{M}$. By grouping the actions, we are able to find the nearly-optimal policy over a reduced-dimensional state-action space, which significantly reduces the complexity. Define the surjective mapping function $g : \mathcal{A} \to \mathcal{G}$, where $g(\boldsymbol{a}) = g(\boldsymbol{a}')$ for any actions $\boldsymbol{a}$ and $\boldsymbol{a}'$ in the same group. The set of actions that belong to group $h$ is denoted as $\mathcal{A}_h$. Define $|g| := |\mathcal{G}|$ as

the number of groups mapped by the grouping function $g$, and $\mathcal{D}$ as the set of all feasible grouping functions.

For each step, we consider a combined policy, where the higher-level policy $\pi^\circ : \boldsymbol{S} \to \Omega(\mathcal{G})$ selects the group and the lower-level policy $\pi^i(\cdot|\boldsymbol{s}, h) : \boldsymbol{S} \times h \to \Omega(\mathcal{A}_h), h \in \mathcal{G}$ selects an action belonging to that group. The joint policy is composed of the higher- and lower-level policies, denoted as $\pi_G = \pi^\circ \circ \pi^{i1}$. The lower-level policy can be obtained using domain knowledge. In the case where actions within the same group exhibit similar transition kernels and reward functions, we can also employ the uniform distribution as the lower-level policy.

To assess the effectiveness of the grouping operation, we introduce a linear decomposition model that quantifies the similarity between actions within the same group based on their transition kernels and rewards. This model allows us to evaluate the extent of the performance degradation, which will be thoroughly discussed in the following sections.

**Grouped transition probability**    Define the linear decomposition of $\mathbb{P}$ by the tuple $(\boldsymbol{\beta}_P, \mathbb{P}_1, \mathbb{P}_2)$ as

$$\mathbb{P}(\boldsymbol{s}'|\boldsymbol{s}, \boldsymbol{a}) = (1 - \boldsymbol{\beta}_P(\boldsymbol{s}, \boldsymbol{a}))\mathbb{P}_1(\boldsymbol{s}'|\boldsymbol{s}, g(\boldsymbol{a})) + \boldsymbol{\beta}_P(\boldsymbol{s}, \boldsymbol{a})\mathbb{P}_2(\boldsymbol{s}'|\boldsymbol{s}, \boldsymbol{a}), \tag{1}$$

where $\boldsymbol{\beta}_P : \boldsymbol{S} \times \boldsymbol{A} \to [0, 1]$, $\mathbb{P}_1 : \mathcal{S} \times \mathcal{G} \to \mathcal{S}$ is the transition probability from the state and group pair belonging to the state space and the group space to the next state, and $\mathbb{P}_2 : \mathcal{S} \times \mathcal{A} \to \mathcal{S}$ is the transition probability from the state and action pair belonging to the state space and the actions space to the next state. Any $\mathbb{P}$ has at least one linear decomposition solution $(\boldsymbol{\beta}_P, \mathbb{P}_1, \mathbb{P}_2)$ of Eq. (1) since there exist a naive linear decomposition solution $(\boldsymbol{\beta}_P(\boldsymbol{s}, \boldsymbol{a}) = 1, \mathbb{P}_1 = \boldsymbol{0}, \mathbb{P}_2 = \mathbb{P})$. Define the probability deviation factor $\beta_P := \max_{\boldsymbol{s}, \boldsymbol{a}} \boldsymbol{\beta}_P(\boldsymbol{s}, \boldsymbol{a})$, thus $0 \leq \beta_P \leq 1$.

**Grouped rewards**    Similar to the transition probability distribution, we write actual rewards $R(\boldsymbol{s}, \boldsymbol{a})$ as the linear combination of $0 \leq R_1(\boldsymbol{s}, \boldsymbol{a}) \leq 1$ and $0 \leq R_2(\boldsymbol{s}, \boldsymbol{a}) \leq 1$ with factor $\boldsymbol{\beta}_R(\boldsymbol{s}, \boldsymbol{a})$, which is shown as

$$R(\boldsymbol{s}, \boldsymbol{a}) = (1 - \boldsymbol{\beta}_R(\boldsymbol{s}, \boldsymbol{a}))R_1(\boldsymbol{s}, g(\boldsymbol{a})) + \boldsymbol{\beta}_R(\boldsymbol{s}, \boldsymbol{a})R_2(\boldsymbol{s}, \boldsymbol{a}). \tag{2}$$

Define the rewards deviation parameter as $\beta_R := \max_{\boldsymbol{s}, \boldsymbol{a}} \boldsymbol{\beta}_R(\boldsymbol{s}, \boldsymbol{a})$ and $0 \leq \beta_R \leq 1$. $R_1$ can be viewed as the reward function corresponding to state-group space, and $R_2$ represents the deviated reward function of the primitive state-action space.

**Remark 1.** *(Obtaining $\mathcal{D}$) Intuitively, actions that have similar transition probability distributions and reward functions can be clustered into the same group. We can use expert knowledge of specific applications to obtain the feasible grouping function set $\mathcal{D}$ before the learning process. Note that we do not make any assumptions on $\mathcal{D}$, e.g., we do not need the finer grouping function to be the refinement of coarser grouping functions as in [Assumption 1, (Jiang et al., 2015)].*

**Remark 2.** *(Calculation of $\mathbb{P}_1$ and $\mathbb{P}_2$) $\mathbb{P}_1$ is the common transition kernel for all actions in the same group, and $\mathbb{P}_2$ reflects each individual action's transition characteristics. We can get $(\mathbb{P}_1, \mathbb{P}_2)$ by either directly solving Eq. (1) or utilizing the domain knowledge. We provide an example of getting $(\mathbb{P}_1, \mathbb{P}_2)$ in the wireless access scenario in Appendix B.1.*

**Remark 3.** *(Meaning of $\beta_P$ and $\beta_R$) The deviation factors $\beta_P$ and $\beta_R$ reflect how well the common transition probability distribution and reward can represent each action's actual transition kernel and reward. If $\beta_P$ and $\beta_R$ are small, then the transition kernel and rewards of joint actions in the same group are almost identical. Specifically, when $\beta_P = \beta_R = 0$, $\mathbb{P}(\cdot|\boldsymbol{s}, \boldsymbol{a}) = \mathbb{P}_1(\cdot|\boldsymbol{s}, g(\boldsymbol{a}))$ and $R(\boldsymbol{s}, \boldsymbol{a}) = R_1(\boldsymbol{s}, g(\boldsymbol{a}))$. When $\beta_P = 1$, $\mathbb{P}(\cdot|\boldsymbol{s}, \boldsymbol{a}) = \mathbb{P}_2(\cdot|\boldsymbol{s}, \boldsymbol{a})$ and there is no common pattern for all actions in the same group.*

### 3.3    MODEL-BASED RL WITH GENERATIVE MODEL

The model-based dynamic programming algorithm with the generative model is shown in Algorithm 1. Assume we can access a generative model that generates independent quadruples $(\boldsymbol{s}, h, r, \boldsymbol{s}')$ following $\mathcal{M}_G$. We generate $K'$ quadruples for each state-group pair, where $r_k = R_G(\boldsymbol{s}, h)$, $\boldsymbol{s}'_k \sim \mathbb{P}_G(\cdot|\boldsymbol{s}, h)$. The total sample complexity is $K = S|g|K'$. We can have an empirical estimation

---

[1]Assume $f : \boldsymbol{S} \to \boldsymbol{A}$, $\phi : \boldsymbol{S} \to \mathcal{G}$, $\psi_h : h \to \mathcal{A}_h$, and $\psi = \{\psi_h\}_{h \in \mathcal{G}}$. We define $f = \phi \circ \psi$ iff $f(\boldsymbol{a}|\boldsymbol{s}) = \phi(h|\boldsymbol{s})\psi_h(\boldsymbol{a}|h)$.

---

**Algorithm 1** Model-based RL with generative model

---

1: **Input:** state value function initialization $\hat{Q}_G^0(s, h) = 0$ for all $s \in \mathcal{S}$, $h \in \mathcal{G}$.
2: **Output:** policy $\pi_T^\circ$.
3: **for** $(s, h) \in \mathcal{S} \times \mathcal{G}$ **do**
4:      Draw sample $(s, h, r_k, s_k')_{k=1}^{K'}$, where $r_k = R_G(s, h)$, $s_k' \sim \mathbb{P}_G(\cdot|s, h)$.
5: **end for**
6: Estimate $\hat{\mathbb{P}}_G$ and $\hat{R}_G$ by Eq. (3).
7: Execute the dynamic programming algorithm for $T$ iterations and generate the policy $\pi_T^\circ$.

---

of $\mathcal{M}$ as

$$\hat{\mathbb{P}}_G(s'|s, h) = \frac{\sum_{k=1}^{K'} \text{count}(s, h, s')}{K'}, \ \hat{R}_G(s, h) = \frac{1}{K'} \sum_{k=1}^{K'} r_k(s, h). \tag{3}$$

We can construct an empirical MDP as $\hat{\mathcal{M}}_G = (\mathcal{S}, \mathcal{A}, \hat{\mathbb{P}}_G, \hat{R}_G, \gamma)$ by sampling over each state-group pair and use the oracle dynamic programming algorithms such as value iteration and policy iteration to get a nearly-optimal policy under the estimated MDP.

Here we consider the sample and computational complexity induced by Algorithm 1. Sample complexity, denoted as $\mathcal{C}_{\text{samp}}$, represents the number of samples needed to obtain the $\epsilon_{\text{perf}}$-optimal policy. On the other hand, computational complexity denoted as $\mathcal{C}_{\text{comp}}$, refers to the computational operations required to achieve the same goal.

## 4 MAIN RESULTS ON PERFORMANCE EVALUATION

We now present the main theorem that establishes the upper bound on the performance gap between the optimal policy and the output policy of Algorithm 1. Let $\pi_{G,T} = \pi_T^\circ \circ \pi^i$, where $\pi_T^\circ$ is the output policy of Algorithm 1 after $T$ iterations.

**Theorem 1.** *Assume the reward is deterministic[2]. Given $\mathcal{M}$ and grouping function $g$, when the value function difference between the optimal policy and the output policy $\pi_T^\circ$ under the estimated MDP $\hat{\mathcal{M}}_G$ denoted as $\left\| V_{\hat{\mathcal{M}}_G}^* - V_{\hat{\mathcal{M}}_G}^{\pi_T^\circ} \right\|_\infty \le \epsilon_{opt}$, and the sample size $K \ge \frac{648S|g|\log\frac{8S|g|}{\delta(1-\gamma)}}{(1-\gamma)^2}$, with probability exceeding $1 - \delta$, one has*

$$\|V_\mathcal{M}^* - V_\mathcal{M}^{\pi_{G,T}}\|_\infty \le \epsilon_{perf},$$

*where*

$$\epsilon_{perf} = \underbrace{2\left(\frac{\gamma\beta_P^*}{(1-\gamma)^2} + \frac{\beta_R^*}{1-\gamma}\right)}_{\text{approximation error } \epsilon_{approx}} + \underbrace{20\gamma\sqrt{\frac{S|g|\log\left(\frac{8S|g|}{\delta(1-\gamma)}\right)}{K(1-\gamma)^3}} + \frac{4\epsilon_{opt}}{1-\gamma}}_{\text{estimation error } \epsilon_{est}}, \tag{4}$$

$$\beta_P^* = 1 - \min_{s \in \mathcal{S}, h \in \mathcal{G}} \sum_{s'} \min_{a \in \mathcal{A}_h} \mathbb{P}(s'|s, a), \tag{5}$$

$$\beta_R^* = \max_{s \in \mathcal{S}, g(a_1) = g(a_2)} (R(s, a_1) - R(s, a_2)). \tag{6}$$

As shown in Eq. (4), the performance gap between $\pi^*$ and $\pi_{G,T}$ contains two part: approximation error and estimation error. We now explain the two error terms as follows.

*Approximation error* arises when the dynamic programming algorithm operates at the group level and ignores the disparities in transition probability distributions and rewards among actions within the same group. Specifically, $\beta_P^*$ and $\beta_R^*$ are the minimal $\beta_P$ and $\beta_R$, where $(\beta_P, \mathbb{P}_1, \mathbb{P}_2)$ and $(\beta_R, R_1, R_2)$ are the solutions to Eqs. (1) and (2), respectively. It is important to note that $\beta_P^*$ and $\beta_R^*$ are solely determined by the grouping function and do not depend on any other factors.

---

[2]This assumption implies $\hat{R}_G$ in Eq. (3) is accurate.

As the number of groups increases, the approximation error generally decreases. The underlying intuition is that a finer grouping function has the potential to improve performance by minimizing grouping errors and capturing subtle distinctions within groups. We illustrate this concept by providing an example, where a finer grouping function in the feasible grouping function set is a refinement of a coarser grouping function. This structure has also been considered in prior work such as Jiang et al. (2015). As the grouping function becomes coarser, the differences in probability transition distribution and rewards within each group become larger, resulting in a monotonic increase in the approximation error.

*Estimation error* can be further decomposed into two terms: $\epsilon_{\text{samp}}$ (the first term of $\epsilon_{\text{est}}$) and $\epsilon_{\text{alg}}$ (the second term of $\epsilon_{\text{est}}$). Specifically, $\epsilon_{\text{samp}}$ and $\epsilon_{\text{alg}}$ reflect the performance loss caused by transition kernel estimation with finite samples and limited iteration numbers, respectively. When using policy iteration (Antos et al., 2008) or value iteration (Munos, 2005; Munos and Szepesvári, 2008) as the planning algorithm in Algorithm 1, $\epsilon_{\text{alg}}$ decreases at a linear rate in the number of iterations $T$. To provide a comprehensive understanding, we present the detailed derivations of $\epsilon_{\text{alg}}$ specifically for the case of value iteration in Appendix D.

The detailed analysis of the tightness of Theorem 1 in special cases and the proof sketch are presented in Section 6.

**Comparison with Wang et al. (2021)** We compare our results with [(Wang et al., 2021), Theorem 3]. Wang et al. (2021) also considers the model-based method utilizing the generative model and investigates the performance loss caused by inaccurate feature extraction in [(Wang et al., 2021), Theorem 3]. We summarize our main differences as follows. Compared with Theorem 1, our result characterizes the performance gap caused by the rewards model deviating from the grouping model, which was not considered in previous work. Furthermore, our result improves the approximation error of transition kernel difference by a factor of $\frac{22}{\gamma}$. We demonstrate the tightness of our approximation error with a constant difference when deviation factors are small enough. Notably, our approach optimizes over $(\mathbb{P}_1, \mathbb{P}_2)$ and $(R_1, R_2)$ to find the minimum $\beta_P^*$ and $\beta_R^*$.

Theorem 1 reveals that the performance loss $\|V_{\mathcal{M}}^* - V_{\mathcal{M}}^{\pi_{G,T}}\|_\infty$ can be mitigated through tuning the sample size, iteration number, and the grouping function. Increasing the sample size $K$ or the number of computational operations $T$ can reduce the performance loss since $K$ and $T$ are inversely related to $\epsilon_{opt}$. This coincides with the intuition that a larger sample dataset and larger available computational complexity leads to a better-performed policy. However, **the relationship between the performance loss and the grouping function is surprising.**

**A grouping function with a larger number of groups can sometimes achieve better performance.** When the sample size and the number of computation operations are extremely large, the estimation error becomes significantly smaller compared to the approximation error, making the performance loss predominantly determined by the approximation error. In such cases, we can choose a grouping function with a larger number of groups to achieve better performance.

As demonstrated in Fig. 1(a), comparing the grouping and non-grouping settings with $K' = 500$ (red and blue lines with marker ○), the non-grouping setting has a smaller estimation error. The number of samples for each action is sufficient for accurate estimation, leading to better performance in the non-grouping setting than in the grouping setting. This observation aligns with our analysis, which suggests the performance loss decrease as the number of groups increases.

**Adjusting grouping function can sometimes enable a trade-off between approximation and estimation errors.** When the sampling size and the computation operations are limited, the estimation error cannot be disregarded. According to Theorem 1, $\epsilon_{\text{samp}}$ decreases sublinearly as the number of groups decreases. This can be intuitively understood: when the sample size is fixed, having fewer groups allows for more samples to estimate the transition probability of each state-group pair, which consequently leads to a smaller estimation error. Similarly, when the computational complexity is limited, a smaller number of groups indicates a larger iteration number $T$, therefore reducing $\epsilon_{\text{alg}}$.

As demonstrated in Fig. 1(a), comparing the grouping and non-grouping settings with $K' = 10$ (red and blue lines with marker □), the non-grouping setting has a higher estimation error than the grouping setting. However, even though the approximation error is smaller in the non-grouping case, setting small deviation factors ensures that the grouping error remains small. Therefore, the

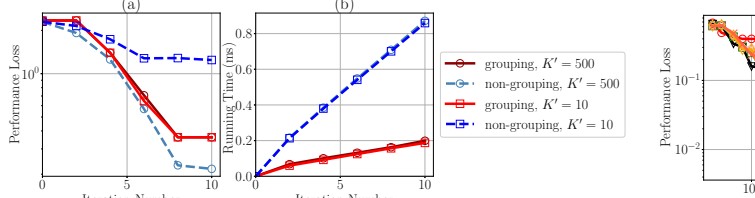 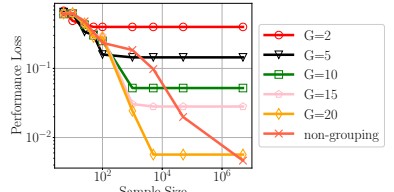

Figure 1: Performance loss with grouping and non-grouping structure under the downlink transmission scenario (details are in Appendix B.2). Each point is averaged over 20 rounds. $A = 1000$, and $G = 10$.

Figure 2: Performance loss versus sample size. Each point is averaged over 1000 rounds. $A = 1000$.

overall performance loss of the non-grouping case is still significantly larger than the grouping setting. Fig. 1(b) implies the grouping can decrease the computational complexity drastically.

This surprising result can also be verified by Fig. 2. When we only have limited sampling resources, coarser grouping functions are preferred. For example, when the total sample size is $K = 10^4$, the grouping function with group number $G = 20$ (yellow line) is preferred over other settings. However, in scenarios where the sample resources are unlimited (e.g., sample size $K = 10^7$), the non-grouping function becomes the preferred option.

The above analysis based on Theorem 1 reveals that optimizing the grouping function is a key factor in reducing performance loss. In Section 5, we propose a general optimization problem considering the trade-off between performance loss and complexity.

## 5 PERFORMANCE-COMPLEXITY TRADE-OFF

Theorem 1 highlights the possibility of optimizing the grouping function to reduce performance loss when all available sample and computational resources are utilized. However, practical applications often involve additional costs for acquiring samples and computational operations, which may not scale proportionally with the size of these resources (Luo et al., 2021). To address this, we aim to find the optimal sample size, iteration number, and grouping function that strike a balance between complexity and performance loss. This trade-off allows us to achieve an acceptable level of performance degradation while keeping the sample and computational complexity manageable.

To capture this trade-off, we introduce a utility function $f : \mathbb{R}^3 \to \mathbb{R}$ that characterizes the preferences for performance loss and complexity. We make the following assumption.

**Assumption 2.** $f$ *is monotone decreasing w.r.t each variable, i.e.,* $f(x_1, y, z) \leq f(x_2, y, z)$ *iff* $x_1 > x_2$, *and same applies to $y$ and $z$.*

Assumption 2 guarantees that the utility function decreases when either the performance loss or the sample/computational complexity increases while holding other terms constant. According to Assumption 2, the minimization of $f(\epsilon_{\text{perf}}, \mathcal{C}_{\text{samp}}, \mathcal{C}_{\text{comp}})$ reflects a preference for lower performance loss and complexity. An example of $f$ is a weighted-sum function: $f(x, y, z) = \alpha_1 x + \alpha_2 y + \alpha_3 z$, where $\boldsymbol{\alpha}$ is a vector of weighting coefficients and $\alpha_i < 0, i = 1, 2, 3$.

The utility-maximization problem can be written as

$$\max_{g \in \mathcal{D}, K, T} f(\epsilon_{\text{perf}}(g, K, T), \mathcal{C}_{\text{samp}}(K), \mathcal{C}_{\text{comp}}(|g|, T)). \tag{P1}$$

Specifically, if we set $f(\epsilon_{\text{perf}}(g, K, T)) = \epsilon_{\text{perf}}(g, K, T)$, and $(\mathcal{C}_{\text{samp}}, \mathcal{C}_{\text{comp}})$ are infinite when exceeding some threshold, the grouping function reduces to the performance loss minimization with fixed sampling complexity and computational complexity. For a given feasible grouping function $g$, the optimization over $K$ and $T$ can be performed independently. Specifically, if $f$ is a convex function of $K$ and $T$, we can directly use convex optimization methods to get the optimal $K^*(|g|)$ and $T^*(|g|)$. The optimization problem can be rewritten as

$$\max_{g \in \mathcal{D}} f(\epsilon_{\text{perf}}(g, K^*(|g|), T^*(|g|)), \mathcal{C}_{\text{samp}}(K^*(|g|)), \mathcal{C}_{\text{comp}}(|g|, T^*(|g|))).$$

By solving the above optimization problem, we can achieve a balance between minimizing performance loss and managing complexity. However, solving the above optimization problem involves several challenges. Firstly, the feasible grouping function set $\mathcal{D}$ is discrete, and $\epsilon_{\text{perf}}$ is implicitly related to the grouping function, therefore the optimization objective is not easily solvable. Additionally, Eq. (5) shows the exact calculation of $\epsilon_{\text{perf}}(g, K^*(|g|), T^*(|g|))$, which requires traversing over all coefficients of $\mathbb{P}$ the $R$, and results in a computational complexity that grows with the size of the action space. Moreover, the exact probability transition kernel is usually not known by the agent. Hence, we next devise a practical approach to handle this problem and analyze its performance.

## 5.1 Practical Method

To mitigate the computational demands of solving (P1), we propose (P2) as its approximated counterpart. Since the computational complexity of solving (P1) is dominated by the calculation of $\beta_P^*$ and $\beta_R^*$ in its objective, we approximate these terms in (P2). Instead of iterating through all the actions within the same group and applying Eqs. (5) and (6) to calculate accurate values of $\beta_P^*$ and $\beta_R^*$, we propose an approximation of $\epsilon_{\text{perf}}(g, K^*(|g|), T^*(|g|))$ based on randomly selected actions. Specifically, we utilize samples obtained from the generative model to estimate the transition probabilities of selected state-action pairs, and then substitute these estimated probabilities into Eq. (4) to obtain the estimation $\hat{\epsilon}_{\text{perf}}$. This approach significantly reduces the computational complexity, as the complexity of the calculation of approximate deviation factors is only related to the group size, which is much smaller than the entire action space. (P1) can be approximated as

$$\max_{g \in \mathcal{D}} f(\hat{\epsilon}_{\text{perf}}(g, K^*(|g|), T^*(|g|)), \mathcal{C}_{\text{samp}}(K^*(|g|)), \mathcal{C}_{\text{comp}}(|g|, T^*(|g|))), \tag{P2}$$

where the calculation of $\hat{\epsilon}_{\text{perf}}(g, K^*(|g|), T^*(|g|))$ only uses actions belonging to $\bar{\mathcal{A}} = \cup_{h \in \mathcal{G}} \bar{\mathcal{A}}_h$, and $\bar{\mathcal{A}}_h$ is the randomly selected actions of the group $h \in \mathcal{G}$. Subsequently, we can iterate through $\mathcal{D}$ to determine the optimal grouping function.

## 5.2 Performance Analysis of the Practical Method

We show that the above approximation in (P2) is reasonable. Intuitively, since actions within a group have comparable transition kernels and reward functions in the grouped action space setting, we can capture intra-group dissimilarity by only selecting a subset from each group. Now We formalize this intuition.

We impose the condition that the rate of change of the utility function remains bounded in response to variations in the performance loss. The following assumptions are presented to capture these requirements.

**Assumption 3.** *($f(x, y, z)$ is Lipschitz continuous w.r.t $x$.) There exists $L > 0$ such that $|f(x_1, y, z) - f(x_2, y, z)| \leq L|x_1 - x_2|$.*

We define $\eta_P$ and $\eta_R$ as

$$
\begin{aligned}
\eta_P &= \max_{\boldsymbol{s}, h, \boldsymbol{a}_1, \boldsymbol{a}_2 \in \mathcal{A}_h} \|\mathbb{P}(\cdot|\boldsymbol{s}, \boldsymbol{a}_1) - \mathbb{P}(\cdot|\boldsymbol{s}, \boldsymbol{a}_2)\|_{\infty}, \\
\eta_R &= \max_{\boldsymbol{s}, h, \boldsymbol{a}_1, \boldsymbol{a}_2 \in \mathcal{A}_h} (R(\boldsymbol{s}, \boldsymbol{a}_1) - R(\boldsymbol{s}, \boldsymbol{a}_2)).
\end{aligned}
\tag{7}
$$

In particular, Eq. (7) represents the proximity of the transition probability distributions and the reward functions in the same group. Note that $\beta_P^* \leq S\eta_P$ and $\beta_R^* \leq \eta_R$.

Denote the optimal grouping function of (P1) as $g^*$ and of (P2) as $\hat{g}^*$, respectively. Correspondingly, the utility functions under $g^*$ and $\hat{g}^*$ are denoted as $f^*$ and $\hat{f}^*$, respectively. Let $K_1$ be the number of total samples required by the estimation of $\epsilon_{\text{perf}}$. We have the following lemma to quantify the gap between $f^*$ and $\hat{f}^*$.

**Proposition 1.** *Assume the reward is deterministic. With probability exceeding $1 - \delta$, we have*

$$f^* - \hat{f}^* \leq \underbrace{\frac{4L\eta_R}{1 - \gamma} + \frac{4L\gamma S\eta_P}{(1 - \gamma)^2}}_{\text{action sampling error}} + \underbrace{\frac{4L\gamma S}{(1 - \gamma)^2} \sqrt{\frac{S|\bar{\mathcal{A}}| \log \frac{2S|\bar{\mathcal{A}}|}{\delta}}{2K_1}}}_{\text{probability estimation error}}.$$

The performance gap between the optimal utility function obtained by solving (P1) and (P2) can be decomposed into two components: the action sampling error and the probability estimation error. In certain MDPs where the actions in the same group are close enough (small $\eta_P$ and $\eta_R$) and the utility function shows limited variation with respect to changes in $\epsilon_{\text{perf}}$ (small $L$), the action sampling error is low. Note that the probability estimation error is associated with the accuracy of estimating the transition probability distribution. The good news is that the required number of samples is only proportional to the number of groups, which is significantly smaller than the size of the entire action space. Therefore, these findings suggest that solving the approximate optimization problem (P2) allows us to obtain the optimal grouping function $g$ with little performance degradation across a wide range of MDPs, while maintaining the sample costs at a reasonable level.

## 6 Proof Sketch of Main Theorem and Tightness Analysis

**Proof Sketch** The presented Theorem 1 establishes the upper bound for $\left\| V_{\mathcal{M}}^* - V_{\mathcal{M}}^{\pi_G,T} \right\|_\infty$, the performance loss between the optimal policy and the policy obtained by our algorithm. Let $\pi_G^*$ be the group-wise optimal policy. We can decompose this difference into two parts: the approximation error $\| V_{\mathcal{M}}^* - V_{\mathcal{M}}^{\pi_G^*} \|_\infty$ and the estimation error $\| V_{\mathcal{M}}^{\pi_G^*} - V_{\mathcal{M}}^{\pi_G,T} \|_\infty$.

To establish an upper bound for the approximation error, we introduce an auxiliary MDP $\mathcal{M}_1 = \{\mathcal{S}, \mathcal{A}, \mathbb{P}_1, R_1, \gamma\}$ which shares the same state and action spaces as the original MDP but differs in terms of transition distribution and rewards. The extent of this dissimilarity can be quantified by parameters $\beta_P$ and $\beta_R$. By comparing the value functions of executing the same policy on $\mathcal{M}_1$ and $\mathcal{M}$, we can derive value function difference in terms of $\beta_P$, $\beta_R$, and the horizon $1/(1-\gamma)$. We can further get the approximation error is upper bounded by the twice of the value function difference under $\mathcal{M}_1$ and $\mathcal{M}$.

To derive the upper bound of the estimation error, we employ the leave-one-out analysis (Agarwal et al., 2020b), which constructs auxiliary MDPs where one state is set as absorbing while the others remain unchanged. This helps us to disentangle the connection between probability kernel estimation $\hat{\mathbb{P}}_G$ and the optimal group-wise policy $\pi_G^*$ (Agarwal et al., 2020b). Compared with (Agarwal et al., 2020b; Wang et al., 2021), we extend the estimation error analysis from tabular MDPs to grouped MDPs, obtaining a minimax optimal upper bound for the estimation error.

**Tightness Analysis** We compare our result with a known lower bound on the estimation error to show that it is tight. Further, we provide an example in Appendix C.3 for which the approximation error is also relatively tight.

*Tightness of Estimation Error:* Recall that the estimation error contains sampling error $\epsilon_{\text{samp}}$ which is related to $K$ and algorithmic error which is related to $T$. While sampling error decreases sublinearly with respect to $K$, the algorithmic error diminishes at a faster, linear rate. Consequently, the limited sample size predominantly limits the estimation performance. The required sample size to achieve $\epsilon$-optimal approximation error is $\tilde{O}(\frac{S|g|}{(1-\gamma)^3 \epsilon^2})$. Due to the lower bound of sample complexity in the generative model (Azar et al., 2012), we achieve the minimax optimal sample complexity.

*Example showing tightness of Approximation Error:* The example is designed such that the group-wise optimal policy $\pi_G^*$ has a high probability of selecting a state-action pair with nearly zero potential reward while letting the optimal policy $\pi^*$ choose state-action pairs that have large potential rewards. We show that for any $\varepsilon > 0$, the difference between the derived performance loss of Theorem 1 and $\epsilon_{\text{approx}}/2$ is smaller than $\varepsilon$ when $\beta_P$ and $\beta_R$ are small enough. This implies that the derived upper bound of the approximation error only differs by a constant factor of 2.

## 7 Conclusion

This paper addresses the curse of dimensionality by exploring the inherent structure of group-wise similar action space. We introduced a linear decomposition model for representing the similarity of actions within the same group. Our work provides insights into the trade-off between complexity and performance loss when applying reinforcement learning algorithms to practical applications.

ACKNOWLEDGMENTS

This work has been supported in part by NSF grants: CNS-2312836, CNS- 2223452, CNS-2225561, CNS-2112471, CNS- 2106933, a grant from the Army Research Office: W911NF-21-1-0244, and was sponsored by the Army Research Laboratory under Cooperative Agreement Number W911NF-23-2-0225. The views and conclusions contained in this document are those of the authors and should not be interpreted as representing the official policies, either expressed or implied, of the Army Research Laboratory or the U.S. Government. The U.S. Government is authorized to reproduce and distribute reprints for Government purposes notwithstanding any copyright notation herein.

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

# A ADDITIONAL NOTATIONS

Define the the set of policies of MDP $\mathcal{M} := (\boldsymbol{S}, \boldsymbol{A}, \mathbb{P}, R, \gamma)$ as $\Pi := \{\pi | \pi : \boldsymbol{S} \to \Omega(\boldsymbol{A})\}$. Given $\mathcal{M}$ and policy $\pi \in \Pi : \boldsymbol{S} \to \Omega(\boldsymbol{A})$, we can have a collection of trajectories $\tau_{\mathcal{M}}^{\pi} = (\boldsymbol{s}_t, \boldsymbol{a}_t, r_t)_{t=0}^{\infty}$ starting from state-action pair $(\boldsymbol{s}_0, \boldsymbol{a}_0)$. Here, $\boldsymbol{s}_{t+1} \sim \mathbb{P}(\cdot | \boldsymbol{s}_t, \boldsymbol{a}_t)$, $a_{t+1} \sim \pi(\cdot | \boldsymbol{s}_{t+1})$, and $r_t = R(\boldsymbol{s}_t, \boldsymbol{a}_t)$. We present the definition of value functions $Q_{\mathcal{M}}$ and $V_{\mathcal{M}}$ as for any $\boldsymbol{s} \in \boldsymbol{S}$ and $\boldsymbol{a} \in \boldsymbol{A}$,

$$Q_{\mathcal{M}}^{\pi}(\boldsymbol{s}, \boldsymbol{a}) := \mathbb{E}_{\tau_{\mathcal{M}}^{\pi}} \left[ \sum_{t=0}^{\infty} \gamma^t r_t \mid \boldsymbol{s}_0 = \boldsymbol{s}, \boldsymbol{a}_0 = \boldsymbol{a} \right], \tag{8}$$

$$V_{\mathcal{M}}^{\pi}(\boldsymbol{s}) := \mathbb{E}_{\boldsymbol{a} \sim \pi(\cdot | \boldsymbol{s})} \left[ Q^{\pi}(\boldsymbol{s}, \boldsymbol{a}) \right], \tag{9}$$

where $\mathbb{E}_{\tau_{\mathcal{M}}^{\pi}}[\cdot]$ is taking expectation with respect to the randomness of $\tau_{\mathcal{M}}^{\pi}$. The optimal policy under $\mathcal{M}$ is defined as $\pi_{\mathcal{M}}^*$, and its corresponding value functions are denoted as $Q_{\mathcal{M}}^*$ and $V_{\mathcal{M}}^*$. Define Bellman optimal operator $\mathcal{T}_{\mathcal{M}} f(\boldsymbol{s}, \boldsymbol{a}) := R(\boldsymbol{s}, \boldsymbol{a}) + \gamma \langle \mathbb{P}(\cdot | \boldsymbol{s}, \boldsymbol{a}), \max_{\boldsymbol{a}'} f(\cdot, \boldsymbol{a}') \rangle$. Bellman optimality equation indicates that $Q_{\mathcal{M}}^*$ is the fixed point of the Bellman optimality operator and satisfies $Q_{\mathcal{M}}^* = \mathcal{T}_{\mathcal{M}} Q_{\mathcal{M}}^*$.

In addition to the set of policies $\Pi$ defined on $\mathcal{M}$, we also define the set of grouped policies based on a "lower-level" policy $\pi^{\mathrm{i}}$ as follows:

$$\Pi_G := \{\pi_G | \pi_G = \pi^{\mathrm{o}} \circ \pi^{\mathrm{i}}, \pi^{\mathrm{o}} : \boldsymbol{S} \to \Omega(\mathcal{G})\}. \tag{10}$$

Let $\pi_G^*$ denote the optimal policy belonging to $\Pi_G$ that maximizes $V_{\mathcal{M}}^{\pi}$.

We also construct some special MDPs based on the grouping function $g$ and "lower-level" policy $\pi^{\mathrm{i}}$. We define the *grouped MDP* as $\mathcal{M}_G := \{\boldsymbol{S}, \mathcal{G}, \mathbb{P}_G, R_G, \gamma\}$, where $\mathbb{P}_G(\boldsymbol{s}' | \boldsymbol{s}, h) := \mathbb{E}_{\boldsymbol{a} \sim \pi^{\mathrm{i}}(\cdot | h)} [\mathbb{P}(\boldsymbol{s}' | \boldsymbol{s}, \boldsymbol{a})]$ and $R_G(\boldsymbol{s}, h) := \mathbb{E}_{\boldsymbol{a} \sim \pi^{\mathrm{i}}(\cdot | h)} [R(\boldsymbol{s}, \boldsymbol{a})]$.

When $\mathbb{P}$ and $R$ of $\mathcal{M}$ have the linear decomposition form as Eqs. (1) and (2), we define $\mathcal{M}_1 := (\boldsymbol{S}, \boldsymbol{A}, \mathbb{P}'_1, R'_1, \gamma)$, where $\mathbb{P}'_1(\boldsymbol{s}' | \boldsymbol{s}, \boldsymbol{a}) := \mathbb{P}_1(\boldsymbol{s}' | \boldsymbol{s}, g(\boldsymbol{a}))$ and $R'_1(\boldsymbol{s}, \boldsymbol{a}) := R_1(\boldsymbol{s}, g(\boldsymbol{a}))$ for any $(\boldsymbol{s}, \boldsymbol{a})$.

For notational simplicity, we write $\boldsymbol{z}_t = (\boldsymbol{s}_t, \boldsymbol{a}_t)$, $\mathbb{P}^{\pi}(\boldsymbol{z}_{t+1} | \boldsymbol{z}_t) = \mathbb{P}(\boldsymbol{s}_{t+1} | \boldsymbol{z}_t) \pi(\boldsymbol{a}_{t+1} | \boldsymbol{s}_{t+1})$, $\mathbb{P}'^{\pi}_1(\boldsymbol{z}_{t+1} | \boldsymbol{z}_t) = \mathbb{P}'_1(\boldsymbol{s}_{t+1} | \boldsymbol{z}_t) \pi(\boldsymbol{a}_{t+1} | \boldsymbol{s}_{t+1})$, and $\mathbb{P}^{\pi}_2(\boldsymbol{z}_{t+1} | \boldsymbol{z}_t) = \mathbb{P}_2(\boldsymbol{s}_{t+1} | \boldsymbol{z}_t) \pi(\boldsymbol{a}_{t+1} | \boldsymbol{s}_{t+1})$.

# B EXAMPLES OF GROUPED DEVIATED MODEL

## B.1 WIRELESS ACCESS EXAMPLE

Consider a wireless access system where multiple users send packets to an access point. Assume we have an access point and $N$ users, where all users generate the same packet arrival rate and possess the same finite queue buffer. The local state of each user is defined as the number of packets in their buffer. When a user's local state is not less than 1, they have two available actions: either remain idle or transmit a packet to the access point. A collision occurs if multiple users attempt to send packets simultaneously, resulting in transmission failure.

The success of a transition in the wireless access system depends not only on collision-free scheduling but also on the system's operational mode since the system may experience failures due to system errors. We denote the probability of the transmitter being in good condition and the corresponding transition kernel under state-action pair $(\boldsymbol{s}, \boldsymbol{a})$ as $\alpha_{\text{good}}(\boldsymbol{s}, \boldsymbol{a})$ and $\mathbb{P}_{\text{good}}(\cdot | \boldsymbol{s})$, respectively. Note that when a collision does not happen, $\mathbb{P}_{\text{good}}$ is irrelevant to which user is sending the packets, that is, $\mathbb{P}_{\text{good}}$ is irrelevant to $\boldsymbol{a}$. We also denote the probability and transition kernel of the system failing due to system error $k$ under the state-action pair $(\boldsymbol{s}, \boldsymbol{a})$ as $\alpha_{\text{err}_k}(\boldsymbol{s}, \boldsymbol{a})$ and $\mathbb{P}_{\text{err}_k}(\cdot | \boldsymbol{s}, \boldsymbol{a})$, respectively.

We can divide all actions into two groups $\mathcal{A}_1$ and $\mathcal{A}_2$ by the grouping function shown as

$$g(\boldsymbol{a}) = \begin{cases} 1, & \text{if } \|\boldsymbol{a}\|_1 = 1; \\ 2, & \text{otherwise.} \end{cases}$$

Let $\beta_P(s,a) = 1 - \alpha_{\text{good}}(s,a)$, $\mathbb{P}_1(\cdot|s,1) = \mathbb{P}_{\text{good}}(\cdot|s)$, and $\mathbb{P}_2(\cdot|s,a) = \frac{\sum_{k=1}^{K} \alpha_{\text{err}_k}(s,a)\mathbb{P}_{\text{err}_k}(\cdot|s,a)}{1-\alpha_{\text{good}}(s,a)}$. For $a \in \mathcal{A}_1$, we can write the transition probability distribution as

$$\mathbb{P}(\cdot|s,a) = \alpha_{\text{good}}(s,a)\mathbb{P}_{\text{good}}(\cdot|s,a) + \sum_{k=1}^{K} \alpha_{\text{err}_k}(s,a)\mathbb{P}_{\text{err}_k}(\cdot|s,a)$$

$$= (1 - \beta_P(s,a))\mathbb{P}_1(\cdot|s,1) + \beta_P(s,a)\mathbb{P}_2(\cdot|s,a).$$

For other actions belonging to $\mathcal{A}_2$, the transmission certainly fails, therefore they have the same transition distribution, which we denote as $\mathbb{P}_{\text{collision}}$. Therefore, for $a \in \mathcal{A}_2$, we have $\mathbb{P}_1(\cdot|s,2) = \mathbb{P}_{\text{collision}}(\cdot|s,a)$, and $\beta_P(s,a) = 0$.

To account for the varying importance levels of users, we assign a reward of $w_{1,n}$ to the user $n$ whose packet is transmitted without collision. Additionally, to discourage long queues, we can introduce a penalty term $\boldsymbol{w}_2^T \boldsymbol{s}$. The reward function is defined as

$$R(\boldsymbol{s},\boldsymbol{a}) = \begin{cases} \max\left(\boldsymbol{w}_1^T \boldsymbol{a} - \boldsymbol{w}_2^T \boldsymbol{s}, 0\right), & \boldsymbol{a} \in \mathcal{A}_1, \\ 0, & \text{otherwise.} \end{cases}$$

### B.2 Downlink wireless transmission

Consider a downlink transmission system where a base station transmits packets to multiple users. In this system, adjacent users typically exhibit similar channel characteristics, leading to similar successful transmission rates. Furthermore, adjacent users often belong to the same category (e.g., machines in a factory) and have similar packet requirements.

Define the state $s \in \{0, 1, \cdots, S-1\}$ as the length of the packet queue at the base station, with a buffer size of $S-1$. The action $a \in \{1, \cdots, A\}$ represents the user with whom the base station establishes a connection. Assume users can be classified into $G$ groups by the grouping function $g : \mathcal{A} \to \{1, \cdots, G\}$, where the set of users belonging to group $h$ is $\mathcal{A}_h$. The arrival rate at the base station is $\lambda$.

We assume the successful transmission rate and reward of user $a \in \mathcal{A}_h$ are $\mu(s,a) = (1 - \tilde{\beta}_P(s,a))\mu_1(s,h) + \tilde{\beta}_P(s,a)w_P(s,a)$ and $R(s,a) = (1 - \tilde{\beta}_R(s,a))R_1(s,h) + \tilde{\beta}_R(s,a)w_R(s,a) - c(s)$, respectively. $\mu_1$ and $R_1$ represent the common transmission rate and common reward function, respectively. $w_P(s,a)$ and $w_R(s,a)$ are independently sampled from the normal distribution. Note that we put penalty $c(s)$ proportional to the queue length to discourage congestion at the base station.

The setting of Fig. 1 and Fig. 2 is as follows. The common successful transmission rate and rewards of all groups are evenly spaced between $[0,1]$, given by $\mu_1(s,h) = \frac{2h-1}{G}$ and $R_1(s,h) = 1 - \frac{2h-1}{G}$. The penalty for queue-length $s$ is defined as $c(s) = \frac{s-1}{S}$. We set $\tilde{\beta}_P(s,a) = \tilde{\beta}_R(s,a) = 0.01$ in Fig. 1 and $\tilde{\beta}_P(s,a) = \tilde{\beta}_R(s,a) = 0.001$ in Fig. 2. The packet arrival rate at the base station is $\lambda = 0.5$. It is worth noting that the transmission rate and reward values are clipped within the range of $[0,1]$. The probability transition matrix can be obtained based on the arrival rate at the base station and the successful transmission rate for each user.

## C Proof of Theorem 1

### C.1 Tightness example

We now give an example to show that the approximation error bound of Theorem 1 is tight with only a constant difference. Consider an MDP $\mathcal{M} = \{\boldsymbol{\mathcal{S}}, \boldsymbol{\mathcal{A}}, \mathbb{P}, R, \gamma, \rho\}$, where $\boldsymbol{\mathcal{S}} = \{s_0, s_1\}$ and $\boldsymbol{\mathcal{A}} = \{a_0, a_1\}$.

The corresponding transition probability and reward function are shown in Table 1. Assume all actions belong to the same group, then it follows that there exist $(\mathbb{P}_1, \mathbb{P}_2)$ and $(R_1, R_2)$ such that $\mathbb{P}$ and $R$ can be decomposed as

$$\mathbb{P}(\cdot|s,a) = (1 - \beta_P)\mathbb{P}_1(\cdot|s) + \beta_P\mathbb{P}_2(\cdot|s,a),$$

$$R(s,a) = (1 - \beta_R)R_1(s) + \beta_R R_2(s,a),$$

| $s'$ \\ $s, a$ | $s_0, a_0$ | $s_0, a_1$ | $s_1, a_0$ | $s_1, a_1$ |
|---|---|---|---|---|
| $s_0$ | 1 | $1 - \beta_P$ | $\beta_P$ | 0 |
| $s_1$ | 0 | $\beta_P$ | $1 - \beta_P$ | 1 |

(a) Transition kernel $\mathbb{P}(s'|s, a)$

| $s$ \\ $a$ | $a_0$ | $a_1$ |
|---|---|---|
| $s_0$ | 0 | $\beta_R$ |
| $s_1$ | $1 - \beta_R$ | 1 |

(b) Reward function $R(s, a)$

Table 1: Transition distribution and rewards setting of the example MDP

| $s'$ \\ $s$ | $s_0$ | $s_1$ |
|---|---|---|
| $s_0$ | 1 | 0 |
| $s_1$ | 0 | 1 |

(a) $\mathbb{P}_1(s'|s)$

| $s'$ \\ $a$ | $s_0/s_1, a_0$ | $s_0/s_1, a_1$ |
|---|---|---|
| $s_0$ | 1 | 0 |
| $s_1$ | 0 | 1 |

(b) $\mathbb{P}_2(s'|s, a)$

| $s$ \\ $a$ | $a_0/a_1$ |
|---|---|
| $s_0$ | 0 |
| $s_1$ | 1 |

(c) $R_1(s)$

| $s$ \\ $a$ | $a_0$ | $a_1$ |
|---|---|---|
| $s_0$ | 0 | 0 |
| $s_1$ | 0 | 1 |

(d) $R_2(s, a)$

Table 2: Linear decomposition of the example MDP

where $\mathbb{P}_1$, $\mathbb{P}_2$, $R_1$, and $R_2$ are shown in Table 2. Set the action-selection policy $\pi^{\mathsf{i}}$ as choosing $a_0$ with probability 1. Therefore, $\pi^*_G = \pi^{\mathsf{o}*} \circ \pi^{\mathsf{i}}$ is

$$\pi^*_G(a|s) = \begin{cases} 1, & a = a_0, s \in \mathcal{S}; \\ 0, & a = a_1, s \in \mathcal{S}. \end{cases}$$

Applying $\pi^*_G$ into the example MDP, we can have the long-term return as

$$V^{\pi^*_G}_{\mathcal{M}}(s_0) = 0,$$
$$V^{\pi^*_G}_{\mathcal{M}}(s_1) = \frac{1 - \beta_R}{1 - (1 - \beta_P)\gamma}.$$

Moreover, it is obvious that the optimal policy that can maximize global long-term returns is

$$\pi^*(a|s) = \begin{cases} 0, & a = a_0, s \in \mathcal{S}; \\ 1, & a = a_1, s \in \mathcal{S}, \end{cases}$$

and the corresponding value function is

$$V^*_{\mathcal{M}}(s_0) = \frac{\beta_R}{1 - \gamma(1 - \beta_P)} + \frac{\beta_P}{(1 - \gamma(1 - \beta_P))(1 - \gamma)},$$
$$V^*_{\mathcal{M}}(s_1) = \frac{1}{1 - \gamma}.$$

Then for any initial state $s \in \mathcal{S}$, we have

$$V^*_{\mathcal{M}}(s) - V^*_{\mathcal{M}_G}(s) = \underbrace{\frac{\beta_R}{1 - (1 - \beta_P)\gamma} + \frac{\gamma\beta_P}{(1 - (1 - \beta_P)\gamma)(1 - \gamma)}}_{\epsilon'_{\text{approx}}}. \tag{11}$$

As shown in Eq. (4) of Theorem 1, the approximation error resulting from treating all actions within the same group as identical is bounded by

$$\epsilon_{\text{approx}} = \frac{2\beta_R}{1 - \gamma} + \frac{2\gamma\beta_P}{(1 - \gamma)^2}, \tag{12}$$

where we replace $\beta^*_P$ and $\beta^*_R$ by $\beta_P$ and $\beta_R$, since we have $\beta^*_P = \beta_P$ and $\beta^*_R = \beta_R$ in this example MDP.

Comparing Eq. (11) and Eq. (12), we can see that the actual error caused by grouping $\epsilon'_{\text{approx}}$ is close to the upper bound $\frac{\epsilon_{\text{approx}}}{2}$ when $\beta_P$ is small.

Specifically, we can show that for any $\varepsilon > 0$, if $\beta_P \le \sqrt{2}(1 - \gamma)^2\varepsilon$ and $\beta_R \le \frac{1-\gamma}{\sqrt{2}\gamma}$, then

$$\left| \frac{\epsilon_{\text{approx}}}{2} - \epsilon'_{\text{approx}} \right| \le \varepsilon.$$

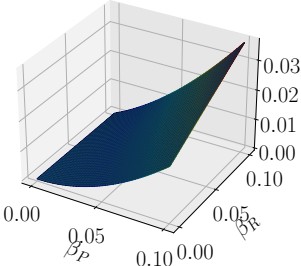

Figure 3: $\left| \frac{\epsilon_{\text{approx}}}{2} - \epsilon'_{\text{approx}} \right|$

To show this, we rewrite $\left| \frac{\epsilon_{\text{approx}}}{2} - \epsilon'_{\text{approx}} \right|$ as

$$\left| \frac{\epsilon_{\text{approx}}}{2} - \epsilon'_{\text{approx}} \right| = \underbrace{\frac{\beta_P \beta_R \gamma}{(1-\gamma)(1-(1-\beta_P)\gamma)}}_{\texttt{a}} + \underbrace{\frac{(\beta_P \gamma)^2}{(1-\gamma)^2 (1-(1-\beta_P)\gamma)}}_{\texttt{b}} \le \varepsilon.$$

By letting $\texttt{a} \le (1-\gamma)\varepsilon$ and $\texttt{b} \le \gamma\varepsilon$, we can get the conditions of $\beta_P$ and $\beta_R$ as

$$\beta_P \le \sqrt{2}(1-\gamma)^2 \varepsilon, \quad \beta_R \le \frac{1-\gamma}{\sqrt{2}\gamma}.$$

We also verify the closeness between $\epsilon'_{\text{approx}}$ and $\frac{\epsilon_{\text{approx}}}{2}$ by simulation. Let $\beta_P$ and $\beta_R$ range from 0 to 0.1 and $\gamma = 0.5$. As shown in Fig. 3, we can approximate $\left\| V^*_{\mathcal{M}} - V^*_{\mathcal{M}_G} \right\|_\infty$ by $\frac{\epsilon_{\text{approx}}}{2}$ when $\beta_P$ and $\beta_R$ are small enough.

Therefore we can achieve the upper bound shown in Lemma 1 with a constant difference.

## C.2    Proof of Theorem 1

The presented Theorem 1 establishes the upper bound for $\left\| V^*_{\mathcal{M}} - V^{\pi_{G,T}}_{\mathcal{M}} \right\|_\infty$, the performance loss between the optimal policy and the policy obtained by our algorithm. The performance loss can be decomposed into two parts: the approximation error $\left\| V^*_{\mathcal{M}} - V^{\pi^*_G}_{\mathcal{M}} \right\|_\infty$ and the estimation error $\left\| V^{\pi^*_G}_{\mathcal{M}} - V^{\pi_{G,T}}_{\mathcal{M}} \right\|_\infty$, where $\pi^*_G$ is the optimal policy belonging to $\Pi_G$. We provide two lemmas to bound these errors: the approximation error lemma, whose derivations can be found in Appendix C.3, and the estimation error lemma, whose derivations can be found in Appendix C.4.

**Lemma 1.** *(Approximation Error Lemma)  Under Assumption 1, we have*

$$\left\| V^*_{\mathcal{M}} - V^{\pi^*_G}_{\mathcal{M}} \right\|_\infty \le 2 \left( \frac{\beta^*_R}{1-\gamma} + \frac{\gamma \beta^*_P}{(1-\gamma)^2} \right),$$

*where $\beta^*_P$ and $\beta^*_R$ are given in Eq. (22).*

**Lemma 2.** *(Estimation Error Lemma)  Assume $\left\| V^*_{\hat{\mathcal{M}}_G} - V^{\pi^\circ_T}_{\hat{\mathcal{M}}_G} \right\|_\infty \le \epsilon_{opt}$. When $K' \ge \frac{648 \log \left( \frac{8S|g|}{\delta(1-\gamma)} \right)}{(1-\gamma)^2}$, with probability larger than $1-\delta$, one has*

$$\left\| V^{\pi^*_G}_{\mathcal{M}} - V^{\pi_{G,T}}_{\mathcal{M}} \right\|_\infty \le 20\gamma \sqrt{\frac{S|g| \log \left( \frac{8S|g|}{\delta(1-\gamma)} \right)}{(1-\gamma)^3 K}} + \frac{4\epsilon_{opt}}{1-\gamma}.$$

Using the above two lemmas, we can easily get the performance loss bound. When $K' \geq \frac{648 \log \left( \frac{8S|g|}{\delta(1-\gamma)} \right)}{(1-\gamma)^2}$, with probability exceeding $1 - \delta$, one has

$$
\left\| V_{\mathcal{M}}^* - V_{\mathcal{M}}^{\pi_{G,T}} \right\|_\infty \leq \left\| V_{\mathcal{M}}^* - V_{\mathcal{M}}^{\pi_G^*} \right\|_\infty + \left\| V_{\mathcal{M}}^{\pi_G^*} - V_{\mathcal{M}}^{\pi_{G,T}} \right\|_\infty \quad \text{(triangle inequality)}
$$

$$
\leq 2 \left( \frac{\beta_R^*}{1 - \gamma} + \frac{\gamma \beta_P^*}{(1 - \gamma)^2} \right) + 20\gamma \sqrt{\frac{S|g| \log \left( \frac{8S|g|}{\delta(1-\gamma)} \right)}{(1 - \gamma)^3 K}} + \frac{4\epsilon_{opt}}{1 - \gamma}
$$

(by Lemma 1 and Lemma 2).

### C.3 Approximation Error: Proof of Lemma 1

*Proof.* We begin by demonstrating that the difference between $V_{\mathcal{M}_1}^*$ and $V_{\mathcal{M}}^*$ is determined by $\beta_P^*$ and $\beta_R^*$. As both $\pi_{\mathcal{M}_1}^*$ and $\pi_G^*$ are policies within $\Pi_G$, with $\pi_G^*$ being the optimal policy in $\Pi_G$, we can subsequently establish an upper bound on the difference between $V_{\mathcal{M}}^{\pi_G^*}$ and $V_{\mathcal{M}}^*$. We now proceed to formalize this derivation.

Define

$$
G_T^\pi(\boldsymbol{z}_0) := \mathbb{E}_{\tau_{\mathcal{M}_1}^\pi} \left[ \sum_{t=0}^T \gamma^t r_t + \gamma^{T+1} \mathbb{E}_{z_{T+1} \sim \mathbb{P}^\pi(\cdot|\boldsymbol{z}_T)} \left[ Q_{\mathcal{M}}^\pi(\boldsymbol{z}_{T+1}) \right] \right]. \tag{13}
$$

Substituting Eq. (1) into $G_T^\pi(\boldsymbol{z}_0)$, we have

$$
\begin{aligned}
G_T^\pi(\boldsymbol{z}_0) =& \mathbb{E}_{\tau_{\mathcal{M}_1}^\pi} \left[ \sum_{t=0}^T \gamma^t r_t + \gamma^{T+1}(1 - \beta_P(\boldsymbol{z}_T)) \mathbb{E}_{\boldsymbol{z}_{T+1} \sim \mathbb{P}_1'^\pi(\cdot|\boldsymbol{z}_T)} \left[ Q_{\mathcal{M}}^\pi(\boldsymbol{z}_{T+1}) \right] \right. \\
& \left. + \gamma^{T+1} \beta_P(\boldsymbol{z}_T) \mathbb{E}_{\boldsymbol{z}_{T+1}' \sim \mathbb{P}_2^\pi(\cdot|\boldsymbol{z}_T)} \left[ Q_{\mathcal{M}}^\pi(\boldsymbol{z}_{T+1}') \right] \right] \\
\leq& \underbrace{\mathbb{E}_{\tau_{\mathcal{M}_1}^\pi} \left[ \sum_{t=0}^T \gamma^t r_t + \gamma^{T+1} \mathbb{E}_{\boldsymbol{z}_{T+1} \sim \mathbb{P}_1'^\pi(\cdot|\boldsymbol{z}_T)} \left[ Q_{\mathcal{M}}^\pi(\boldsymbol{z}_{T+1}) \right] \right]}_{A_1} \\
& + \beta_P \gamma^{T+1} \mathbb{E}_{\tau_{\mathcal{M}_1}^\pi} \left[ \mathbb{E}_{\boldsymbol{z}_{T+1}' \sim \mathbb{P}_2^\pi(\cdot|\boldsymbol{z}_T)} \left[ Q_{\mathcal{M}}^\pi(\boldsymbol{z}_{T+1}') \right] \right] \quad \text{(using } 0 \leq \beta_P(\boldsymbol{z}_T) \leq \beta_P) \\
\leq& A_1 + \frac{\beta_P \gamma^{T+1}}{1 - \gamma} \quad \text{(by } Q_{\mathcal{M}}^\pi(\boldsymbol{z}) \leq \frac{1}{1 - \gamma}).
\end{aligned} \tag{14}
$$

We then derive the upper bound of $A_1$. Since

$$
Q_{\mathcal{M}}(\boldsymbol{z}_t) = \mathbb{E}_{r_t \sim R(\boldsymbol{z}_t)} \left[ r_t \right] + \gamma \mathbb{E}_{\boldsymbol{z}_{t+1} \sim \mathbb{P}^\pi(\cdot|\boldsymbol{z}_t)} \left[ Q_{\mathcal{M}}^\pi(\boldsymbol{z}_{t+1}) \right],
$$

we have

$$
\begin{aligned}
A_1 =& \mathbb{E}_{\tau^\pi_{\mathcal{M}_1}}\left[\sum_{t=0}^{T}\gamma^t r_t + \gamma^{T+1}\mathbb{E}_{\boldsymbol{z}_{T+1}\sim\mathbb{P}'^\pi_1(\cdot|\boldsymbol{z}_T)}\left[\mathbb{E}_{r_{T+1}\sim R(\boldsymbol{z}_{T+1})}\left[r_{T+1}\right.\right.\right.\\
&\left.\left.\left.+\gamma\mathbb{E}_{\boldsymbol{z}_{T+2}\sim\mathbb{P}^\pi(\cdot|\boldsymbol{z}_{T+1})}\left[Q^\pi_\mathcal{M}(\boldsymbol{z}_{T+2})\right]\right]\right]\right]\\
=& \mathbb{E}_{\tau^\pi_{\mathcal{M}_1}}\left[\sum_{t=0}^{T}\gamma^t r_t + \gamma^{T+1}\mathbb{E}_{\boldsymbol{z}_{T+1}\sim\mathbb{P}'^\pi_1(\cdot|\boldsymbol{z}_T)}\left[(1-\beta_R(\boldsymbol{z}_{T+1}))\mathbb{E}_{r_{T+1}\sim R'_1(\boldsymbol{z}_{T+1})}\left[r_{T+1}\right]\right.\right.\\
&\left.\left.+\beta_R(\boldsymbol{z}_{T+1})\mathbb{E}_{r_{T+1}\sim R_2(\boldsymbol{z}_{T+1})}\left[r_{T+1}\right]+\gamma\mathbb{E}_{\boldsymbol{z}_{T+2}\sim\mathbb{P}^\pi(\cdot|\boldsymbol{z}_{T+1})}\left[Q^\pi_\mathcal{M}(\boldsymbol{z}_{T+2})\right]\right]\right] \text{ (by Eq. (2))}\\
=& \mathbb{E}_{\tau^\pi_{\mathcal{M}_1}}\left[\sum_{t=0}^{T}\gamma^t r_t + \gamma^{T+1}\mathbb{E}_{\boldsymbol{z}_{T+1}\sim\mathbb{P}'^\pi_1(\cdot|\boldsymbol{z}_T)}\left[\mathbb{E}_{r_{T+1}\sim R'_1(\boldsymbol{z}_{T+1})}\left[r_{T+1}\right]\right.\right.\\
&\left.\left.+\beta_R\mathbb{E}_{r_{T+1}\sim R_2(\boldsymbol{z}_{T+1})}\left[r_{T+1}\right]+\gamma\mathbb{E}_{\boldsymbol{z}_{T+2}\sim\mathbb{P}^\pi(\cdot|\boldsymbol{z}_{T+1})}\left[Q^\pi_\mathcal{M}(\boldsymbol{z}_{T+2})\right]\right]\right] \text{ (by } 0\le\beta_R(\boldsymbol{z}_T)\le\beta_P)\\
=& \mathbb{E}_{\tau^\pi_{\mathcal{M}_1}}\left[\sum_{t=0}^{T+1}\gamma^t r_t + \gamma\mathbb{E}_{\boldsymbol{z}_{T+2}\sim\mathbb{P}^\pi(\cdot|\boldsymbol{z}_{T+1})}\left[Q^\pi_\mathcal{M}(\boldsymbol{z}_{T+2})\right]\right] + \beta_R\gamma^{T+1} \quad \text{(by } r_{T+1}\le 1)\\
=& G^\pi_{T+1}(\boldsymbol{z}_0) + \beta_R\gamma^{T+1}.
\end{aligned}
\tag{15}
$$

Plugging Eq. (14) into Eq. (15), we have

$$
G^\pi_T(\boldsymbol{z}_0) \le G^\pi_{T+1}(\boldsymbol{z}_0) + \beta_R\gamma^{T+1} + \frac{\beta_P\gamma^{T+1}}{1-\gamma}.
\tag{16}
$$

We can expand $Q^\pi_\mathcal{M}(\boldsymbol{z}_0)$ as

$$
\begin{aligned}
Q^\pi_\mathcal{M}(\boldsymbol{z}_0) =& \mathbb{E}_{r_0\sim R(\boldsymbol{z}_0)}\left[r_0\right] + \mathbb{E}_{\boldsymbol{z}_1\sim\mathbb{P}^\pi(\cdot|\boldsymbol{z}_0)}\left[Q^\pi_\mathcal{M}(\boldsymbol{z}_1)\right]\\
\le& \mathbb{E}_{r_0\sim R'_1(\boldsymbol{z}_0)}\left[r_0\right] + \beta_R\mathbb{E}_{r_0\sim R_2(\boldsymbol{z}_0)}\left[r_0\right] + \mathbb{E}_{\boldsymbol{z}_1\sim\mathbb{P}^\pi(\cdot|\boldsymbol{z}_0)}\left[Q^\pi_\mathcal{M}(\boldsymbol{z}_1)\right] \quad \text{(by Eq. (2))}\\
=& \mathbb{E}_{\tau^\pi_{\mathcal{M}_1}}\left[r_0 + \gamma\mathbb{E}_{\boldsymbol{z}_1\sim\mathbb{P}^\pi(\cdot|\boldsymbol{z}_0)}\left[Q^\pi_\mathcal{M}(\boldsymbol{z}_1)\right]\right] + \beta_R\\
=& G^\pi_0(\boldsymbol{z}_0) + \beta_R \quad \text{(by definition of } G^\pi_0(\boldsymbol{z}_0) \text{ in Eq. (13))}\\
\le& \lim_{T\to\infty}\left[G^\pi_T(\boldsymbol{z}_0) + \sum_{t=0}^{T+1}\beta_R\gamma^t + \sum_{t=1}^{T+1}\frac{\beta_P\gamma^t}{1-\gamma}\right]\\
& \text{(by applying Eq. (16) for } T \text{ times and letting } T \text{ approach infinity)}\\
=& \lim_{T\to\infty}G^\pi_T(\boldsymbol{z}_0) + \frac{\beta_R}{1-\gamma} + \frac{\beta_P\gamma}{(1-\gamma)^2}.
\end{aligned}
\tag{17}
$$

Specifically,

$$
\begin{aligned}
\lim_{T\to\infty}G^\pi_T(\boldsymbol{z}_0) =& \lim_{T\to\infty}\mathbb{E}_{\tau^\pi_{\mathcal{M}_1}}\left[\sum_{t=0}^{T}\gamma^t r_t + \gamma^{T+1}\mathbb{E}_{\boldsymbol{z}_{T+1}\sim\mathbb{P}^\pi(\cdot|\boldsymbol{z}_T)}\left[Q^\pi_\mathcal{M}(\boldsymbol{z}_{T+1})\right]\right]\\
=& \lim_{T\to\infty}\mathbb{E}_{\tau^\pi_{\mathcal{M}_1}}\left[\sum_{t=0}^{T}\gamma^t r_t\right] = Q^\pi_{\mathcal{M}_1}(\boldsymbol{z}_0).
\end{aligned}
\tag{18}
$$

Plugging Eq. (18) into Eq. (17), we finally get

$$
\begin{aligned}
\left\|Q^\pi_\mathcal{M} - Q^\pi_{\mathcal{M}_1}\right\|_\infty =& \max_{\boldsymbol{z}}\left|Q^\pi_\mathcal{M}(\boldsymbol{z}) - Q^\pi_{\mathcal{M}_1}(\boldsymbol{z})\right|\\
\le& \frac{\beta_R}{1-\gamma} + \frac{\beta_P\gamma}{(1-\gamma)^2}.
\end{aligned}
\tag{19}
$$

To get a tighter bound, we want to get the minimum $\beta_P^*$ and $\beta_R^*$ such that there exist $(\beta_P^*, \mathbb{P}_1^*, \mathbb{P}_2^*)$ and $(\beta_R^*, R_1^*, R_2^*)$ satisfying Eq. (1) and Eq. (2). The $\beta_P$ minimization problem can be written as

$$
\begin{aligned}
\min \quad & \beta_P \\
\text{s.t.} \quad & \text{Eq. (1)}, && (s', s, a) \in \boldsymbol{\mathcal{S}} \times \boldsymbol{\mathcal{S}} \times \boldsymbol{\mathcal{A}}, \\
& \textstyle\sum_{s' \in \mathcal{S}} \mathbb{P}_1(s'|s, h) = 1, && (s, h) \in \boldsymbol{\mathcal{S}} \times \mathcal{G}, \\
& \textstyle\sum_{s' \in \mathcal{S}} \mathbb{P}_2(s'|s, a) = 1, && (s, a) \in \boldsymbol{\mathcal{S}} \times \boldsymbol{\mathcal{A}}, \\
& \mathbb{P}_1(s'|s, h) \geq 0, && (s', s, h) \in \boldsymbol{\mathcal{S}} \times \boldsymbol{\mathcal{S}} \times \mathcal{G}, \\
& \mathbb{P}_2(s'|s, a) \geq 0, && (s', s, a) \in \boldsymbol{\mathcal{S}} \times \boldsymbol{\mathcal{S}} \times \boldsymbol{\mathcal{A}}, \\
& 0 \leq \boldsymbol{\beta}_P(s, a) \leq \beta_P, && (s, a) \in \boldsymbol{\mathcal{S}} \times \boldsymbol{\mathcal{A}},
\end{aligned}
\tag{20}
$$

where Eq. (1) is the linear decomposition of $\mathbb{P}$, the last constraint is the the definition of $\beta_P$, and all other constraints ensure $\mathbb{P}_1$ and $\mathbb{P}_2$ are transition probability distributions.

The $\beta_R$ minimization problem can be built in a similar way as

$$
\begin{aligned}
\min \quad & \beta_R \\
\text{s.t.} \quad & \text{Eq. (2)}, && (s, a) \in \boldsymbol{\mathcal{S}} \times \boldsymbol{\mathcal{A}}, \\
& 0 \leq R_1(s, h) \leq 1, && (s, h) \in \boldsymbol{\mathcal{S}} \times \mathcal{G}, \\
& 0 \leq R_2(s, a) \leq 1, && (s, a) \in \boldsymbol{\mathcal{S}} \times \boldsymbol{\mathcal{A}}, \\
& 0 \leq \boldsymbol{\beta}_R(s, a) \leq \beta_R, && (s, a) \in \boldsymbol{\mathcal{S}} \times \boldsymbol{\mathcal{A}}.
\end{aligned}
\tag{21}
$$

The derivations presented in Appendix C.5.1 reveal the following minimum values for $\beta_P$ and $\beta_R$.

$$
\begin{aligned}
\beta_P^* &= \max_{s \in \boldsymbol{\mathcal{S}}, h \in \mathcal{G}} \left(1 - \sum_{s' \in \boldsymbol{\mathcal{S}}} \min_{a \in \mathcal{A}_h} \mathbb{P}(s'|s, a)\right), \\
\beta_R^* &= \max_{s \in \boldsymbol{\mathcal{S}}, h \in \mathcal{G}} \left(\max_{a \in \mathcal{A}_h} R(s, a) - \min_{a \in \mathcal{A}_h} R(s, a)\right).
\end{aligned}
\tag{22}
$$

Plugging $\beta_P^*$ and $\beta_R^*$ into Eq. (19), we have

$$
\left\| Q_{\mathcal{M}}^\pi - Q_{\mathcal{M}_1}^\pi \right\|_\infty \leq \frac{\beta_R^*}{1 - \gamma} + \frac{\beta_P^* \gamma}{(1 - \gamma)^2}.
\tag{23}
$$

We can write $V_{\mathcal{M}}^*(s) - V_{\mathcal{M}}^{\pi_{\mathcal{M}_1}^*}(s)$ as

$$
\begin{aligned}
V_{\mathcal{M}}^*(s) - V_{\mathcal{M}}^{\pi_{\mathcal{M}_1}^*}(s) =& V_{\mathcal{M}}^*(s) + (-V_{\mathcal{M}_1}^{\pi_{\mathcal{M}}^*}(s) + V_{\mathcal{M}_1}^{\pi_{\mathcal{M}}^*}(s)) + (-V_{\mathcal{M}_1}^*(s) + V_{\mathcal{M}_1}^*(s)) - V_{\mathcal{M}}^{\pi_{\mathcal{M}_1}^*}(s) \\
=& (V_{\mathcal{M}}^*(s) - V_{\mathcal{M}_1}^{\pi_{\mathcal{M}}^*}(s)) + (V_{\mathcal{M}_1}^{\pi_{\mathcal{M}}^*}(s) - V_{\mathcal{M}_1}^*(s)) + (V_{\mathcal{M}_1}^*(s) - V_{\mathcal{M}}^{\pi_{\mathcal{M}_1}^*}(s)) \\
\leq& (V_{\mathcal{M}}^*(s) - V_{\mathcal{M}_1}^{\pi_{\mathcal{M}}^*}(s)) + (V_{\mathcal{M}_1}^*(s) - V_{\mathcal{M}}^{\pi_{\mathcal{M}_1}^*}(s)) \text{ (by } V_{\mathcal{M}_1}^{\pi_{\mathcal{M}}^*}(s) - V_{\mathcal{M}_1}^*(s) \leq 0) .
\end{aligned}
$$

Since $V_{\mathcal{M}}^*(s) - V_{\mathcal{M}}^{\pi_{\mathcal{M}_1}^*}(s) \geq 0$, we have

$$
\begin{aligned}
\left\| V_{\mathcal{M}}^* - V_{\mathcal{M}}^{\pi_{\mathcal{M}_1}^*} \right\|_\infty &\leq \left\| V_{\mathcal{M}}^* - V_{\mathcal{M}_1}^{\pi_{\mathcal{M}}^*} + V_{\mathcal{M}_1}^* - V_{\mathcal{M}}^{\pi_{\mathcal{M}_1}^*} \right\|_\infty \\
&\leq \left\| V_{\mathcal{M}}^* - V_{\mathcal{M}_1}^{\pi_{\mathcal{M}}^*} \right\|_\infty + \left\| V_{\mathcal{M}_1}^* - V_{\mathcal{M}}^{\pi_{\mathcal{M}_1}^*} \right\|_\infty && \text{(by triangle inequality)} \\
&= \max_s \left| \mathbb{E}_{a \sim \pi_{\mathcal{M}}^*(\cdot|s)} \left[ Q_{\mathcal{M}}^*(s, a) - Q_{\mathcal{M}_1}^{\pi_{\mathcal{M}}^*}(s, a) \right] \right| \\
&\quad + \max_s \left| \mathbb{E}_{a \sim \pi_{\mathcal{M}_1}^*(\cdot|s)} \left[ Q_{\mathcal{M}_1}^*(s, a) - Q_{\mathcal{M}}^{\pi_{\mathcal{M}_1}^*}(s, a) \right] \right| \\
&\leq \max_{s, a} \left| Q_{\mathcal{M}}^*(s, a) - Q_{\mathcal{M}_1}^{\pi_{\mathcal{M}}^*}(s, a) \right| + \max_{s, a} \left| Q_{\mathcal{M}_1}^*(s, a) - Q_{\mathcal{M}}^{\pi_{\mathcal{M}_1}^*}(s, a) \right| \\
&= \left\| Q_{\mathcal{M}}^* - Q_{\mathcal{M}_1}^{\pi_{\mathcal{M}}^*} \right\|_\infty + \left\| Q_{\mathcal{M}_1}^* - Q_{\mathcal{M}}^{\pi_{\mathcal{M}_1}^*} \right\|_\infty \\
&\leq 2 \left( \frac{\beta_R^*}{1 - \gamma} + \frac{\gamma \beta_P^*}{(1 - \gamma)^2} \right).
\end{aligned}
\tag{24}
$$

We can write $V_{\mathcal{M}}^*(s) - V_{\mathcal{M}}^{\pi_G^*}(s)$ as

$$
\begin{aligned}
V_{\mathcal{M}}^*(s) - V_{\mathcal{M}}^{\pi_G^*}(s) =& V_{\mathcal{M}}^{\pi^*}(s) + (-V_{\mathcal{M}}^{\pi_{\mathcal{M}_1}^*}(s) + V_{\mathcal{M}}^{\pi_{\mathcal{M}_1}^*}(s)) - V_{\mathcal{M}}^{\pi_G^*}(s) \\
\leq& V_{\mathcal{M}}^{\pi^*}(s) - V_{\mathcal{M}}^{\pi_{\mathcal{M}_1}^*}(s) \\
& (\text{by } \pi_{\mathcal{M}_1}^* \in \Pi_G \text{ and } \pi_G^* = \arg\max_{\pi \in \Pi_G} V_{\mathcal{M}}^{\pi}).
\end{aligned}
$$

Since $V_{\mathcal{M}}^*(s) - V_{\mathcal{M}_G}^*(s) \geq 0$ for all $s \in \mathcal{S}$, we can apply infinity norm to both sides of the above equation and get

$$
\begin{aligned}
\left\| V_{\mathcal{M}}^* - V_{\mathcal{M}_G}^* \right\|_\infty \leq& \left\| V_{\mathcal{M}}^{\pi^*} - V_{\mathcal{M}}^{\pi_{\mathcal{M}_1}^*} \right\|_\infty \\
\leq& 2\left( \frac{\beta_R^*}{1-\gamma} + \frac{\gamma\beta_P^*}{(1-\gamma)^2} \right) \quad (\text{by Eq. (24)}).
\end{aligned}
$$

This concludes the proof of Lemma 1. $\qquad\qquad\qquad\qquad\qquad\qquad\qquad\qquad\square$

### C.4 ESTIMATION ERROR: PROOF OF LEMMA 2

*Proof.* To quantify the estimation error, we convert the upper bound to the disparity in policy performance between two variants: $\mathcal{M}_G$ and $\hat{\mathcal{M}}_G$. To accomplish this, we utilize the leave-one-out analysis (Agarwal et al., 2020b). In this analysis, we create an auxiliary MDP denoted as $\hat{\mathcal{M}}_{G,s,u}$, where one state $s$ is treated as an absorbing state while leaving the rest unchanged. This approach enables us to disentangle the relationship between the estimation of the probability kernel $\hat{\mathbb{P}}_G$ and the optimal policy for the entire group $\pi_G^*$.

We can write $V_{\mathcal{M}_G}^*(s) - V_{\mathcal{M}_G}^{\pi_T^\circ}(s)$ as

$$
\begin{aligned}
V_{\mathcal{M}_G}^*(s) - V_{\mathcal{M}_G}^{\pi_T^\circ}(s) =& V_{\mathcal{M}_G}^*(s) + (-V_{\hat{\mathcal{M}}_G}^{\pi^{\circ*}}(s) + V_{\hat{\mathcal{M}}_G}^{\pi^{\circ*}}(s)) + (-V_{\hat{\mathcal{M}}_G}^*(s) + V_{\hat{\mathcal{M}}_G}^*(s)) \\
& + (-V_{\hat{\mathcal{M}}_G}^{\pi_T^\circ}(s) + V_{\hat{\mathcal{M}}_G}^{\pi_T^\circ}(s)) - V_{\mathcal{M}_G}^{\pi_T^\circ}(s) \\
\leq& (V_{\mathcal{M}_G}^*(s) - V_{\hat{\mathcal{M}}_G}^{\pi^{\circ*}}(s)) + (V_{\hat{\mathcal{M}}_G}^*(s) - V_{\hat{\mathcal{M}}_G}^{\pi_T^\circ}(s)) + (V_{\hat{\mathcal{M}}_G}^{\pi_T^\circ}(s) - V_{\mathcal{M}_G}^{\pi_T^\circ}(s)) \\
& (\text{by } V_{\hat{\mathcal{M}}_G}^{\pi^{\circ*}}(s) \leq V_{\hat{\mathcal{M}}_G}^*(s) \text{ for any } s).
\end{aligned}
\tag{25}
$$

Applying infinite norm on both sides, we have

$$
\begin{aligned}
\left\| V_{\mathcal{M}_G}^* - V_{\mathcal{M}_G}^{\pi_T^\circ} \right\|_\infty \leq& \left\| V_{\mathcal{M}_G}^* - V_{\hat{\mathcal{M}}_G}^{\pi^{\circ*}} \right\|_\infty + \left\| V_{\hat{\mathcal{M}}_G}^* - V_{\hat{\mathcal{M}}_G}^{\pi_T^\circ} \right\|_\infty + \left\| V_{\hat{\mathcal{M}}_G}^{\pi_T^\circ} - V_{\mathcal{M}_G}^{\pi_T^\circ} \right\|_\infty \\
\leq& \left\| Q_{\mathcal{M}_G}^* - Q_{\hat{\mathcal{M}}_G}^{\pi^{\circ*}} \right\|_\infty + \epsilon_{opt} + \left\| Q_{\hat{\mathcal{M}}_G}^{\pi_T^\circ} - Q_{\mathcal{M}_G}^{\pi_T^\circ} \right\|_\infty \\
& (\text{by } \left\| V_{\hat{\mathcal{M}}_G}^* - V_{\hat{\mathcal{M}}_G}^{\pi_T^\circ} \right\|_\infty \leq \epsilon_{opt}).
\end{aligned}
\tag{26}
$$

We first bound $\left\| Q_{\hat{\mathcal{M}}_G}^{\pi_T^\circ} - Q_{\mathcal{M}_G}^{\pi_T^\circ} \right\|_\infty$, and $\left\| Q_{\mathcal{M}_G}^* - Q_{\hat{\mathcal{M}}_G}^{\pi^{\circ*}} \right\|_\infty$ can then be bounded in the same manner.

By Lemma 4, we have

$$
\left\| Q_{\hat{\mathcal{M}}_G}^{\pi_T^\circ} - Q_{\mathcal{M}_G}^{\pi_T^\circ} \right\|_\infty \leq \left\| \gamma(\mathbf{I} - \gamma\mathbb{P}_G^{\pi_T^\circ})^{-1}(\hat{\mathbb{P}}_G - \mathbb{P}_G)V_{\hat{\mathcal{M}}_G}^* \right\|_\infty + \frac{\gamma\epsilon_{opt}}{1-\gamma}.
\tag{27}
$$

Since $V_{\hat{\mathcal{M}}_G}^*$ is not independent with $\hat{\mathbb{P}}_G$, therefore we cannot directly use the concentration of the sum of independent variables to bound the sampling error $\left| (\hat{\mathbb{P}}_G - \mathbb{P}_G)V_{\hat{\mathcal{M}}_G}^* \right|$. We construct the $s$-absorbing MDP $\hat{\mathcal{M}}_{G,s,u} = \{\mathcal{S}, \mathcal{G}, \hat{\mathbb{P}}_{G,s,u}, \hat{R}_{G,s,u}, \gamma\}$, where $s \in \mathcal{S}$, $u \in \mathcal{U}_s$, and $\mathcal{U}_s$ is the set of the feasible $u$ in state $s$. For any $h_0 \in \mathcal{G}$ and $s_0 \in \mathcal{S}$, $\hat{\mathbb{P}}_{G,s,u}(\cdot|s_0,h_0)$ and $\hat{R}_{G,s,u}(s_0,h_0)$ are

defined as

$$\hat{\mathbb{P}}_{G,\boldsymbol{s},u}(\boldsymbol{s}_1|\boldsymbol{s}_0,h_0) := \begin{cases} \hat{\mathbb{P}}_G(\boldsymbol{s}_1|\boldsymbol{s}_0,h_0), & \boldsymbol{s}_0 \neq \boldsymbol{s}, \boldsymbol{s}_1 \in \boldsymbol{\mathcal{S}}, \\ 1, & \boldsymbol{s}_0 = \boldsymbol{s}, \boldsymbol{s}_1 = \boldsymbol{s}, \\ 0, & \boldsymbol{s}_0 = \boldsymbol{s}, \boldsymbol{s}_1 \neq \boldsymbol{s}, \end{cases}$$

$$\hat{R}_{G,\boldsymbol{s},u}(\boldsymbol{s}_0,h_0) := \begin{cases} R_G(\boldsymbol{s}_0,h_0), & \boldsymbol{s}_0 \neq \boldsymbol{s}, \\ u, & \boldsymbol{s}_0 = \boldsymbol{s}. \end{cases}$$

Denote $(\hat{\mathbb{P}}_G)_{(\boldsymbol{s},h)} = \hat{\mathbb{P}}_G(\cdot|\boldsymbol{s},h)$. We can rewrite $\left| (\hat{\mathbb{P}}_G - \mathbb{P}_G) V^*_{\hat{\mathcal{M}}_G} \right|$ as

$$\left| (\hat{\mathbb{P}}_G - \mathbb{P}_G)_{(\boldsymbol{s},h)} V^*_{\hat{\mathcal{M}}_G} \right| \leq \left| (\hat{\mathbb{P}}_G - \mathbb{P}_G)_{(\boldsymbol{s},h)} V^*_{\hat{\mathcal{M}}_{G,s,u}} \right| + \left| (\hat{\mathbb{P}}_G - \mathbb{P}_G)_{(\boldsymbol{s},h)} \left( V^*_{\hat{\mathcal{M}}_{G,s,u}} - V^*_{\hat{\mathcal{M}}_G} \right) \right|$$

$$\leq \left| (\hat{\mathbb{P}}_G - \mathbb{P}_G)_{(\boldsymbol{s},h)} V^*_{\hat{\mathcal{M}}_{G,s,u}} \right| + \left\| V^*_{\hat{\mathcal{M}}_{G,s,u}} - V^*_{\hat{\mathcal{M}}_G} \right\|_\infty \tag{28}$$

$$\text{(since } \left\| (\hat{\mathbb{P}}_G - \mathbb{P}_G)_{(\boldsymbol{s},h)} \right\|_1 \leq 1 ).$$

Note that $V^*_{\hat{\mathcal{M}}_{G,s,u}}$ is independent with $(\hat{\mathbb{P}}_G)_{(\boldsymbol{s},h)}$ and $(\mathbb{P}_G)_{(\boldsymbol{s},h)}$. We can use the variant of Bernstein's inequality Lemma 7 and union bound to bound the first term. With probability exceeding $1 - \delta/2$, for all $\boldsymbol{s}, h, u$, one has

$$\left| (\hat{\mathbb{P}}_G - \mathbb{P}_G)_{(\boldsymbol{s},h)} V^*_{\hat{\mathcal{M}}_{G,s,u}} \right| \leq \frac{4 \log 4|\mathcal{U}_{\boldsymbol{s}}||S||g|/\delta}{3K'(1-\gamma)} + \sqrt{\frac{4 \log 4|\mathcal{U}_{\boldsymbol{s}}||S||g|/\delta}{K'} \mathrm{Var}_{(\mathbb{P}_G)_{(\boldsymbol{s},h)}} \left[ V^*_{\hat{\mathcal{M}}_{G,s,u}} \right]}$$

$$\overset{(a)}{\leq} \frac{4 \log 4|\mathcal{U}_{\boldsymbol{s}}||S||g|/\delta}{3K'(1-\gamma)} + \sqrt{\frac{4 \log 4|\mathcal{U}_{\boldsymbol{s}}||S||g|/\delta}{K'}} \left\| V^*_{\hat{\mathcal{M}}_{G,s,u}} - V^*_{\hat{\mathcal{M}}_G} \right\|_\infty$$

$$+ \sqrt{\frac{4 \log 4|\mathcal{U}_{\boldsymbol{s}}||S||g|/\delta}{K'}} \sqrt{\mathrm{Var}_{(\mathbb{P}_G)_{(\boldsymbol{s},h)}} \left[ V^*_{\hat{\mathcal{M}}_G} \right]}, \tag{29}$$

where $(a)$ is because

$$\sqrt{\mathrm{Var}_{(\mathbb{P}_G)_{(\boldsymbol{s},h)}} \left[ V^*_{\hat{\mathcal{M}}_{G,s,u}} \right]} \leq \sqrt{\mathrm{Var}_{(\mathbb{P}_G)_{(\boldsymbol{s},h)}} \left[ V^*_{\hat{\mathcal{M}}_{G,s,u}} - V^*_{\hat{\mathcal{M}}_G} \right]} + \sqrt{\mathrm{Var}_{(\mathbb{P}_G)_{(\boldsymbol{s},h)}} \left[ V^*_{\hat{\mathcal{M}}_G} \right]}$$

$$\text{(by } \sqrt{\mathrm{Var}_{\mathbb{P}}[X+Y]} \leq \sqrt{\mathrm{Var}_{\mathbb{P}}[X]} + \sqrt{\mathrm{Var}_{\mathbb{P}}[Y]} )$$

$$\leq \left\| V^*_{\hat{\mathcal{M}}_{G,s,u}} - V^*_{\hat{\mathcal{M}}_G} \right\|_\infty + \sqrt{\mathrm{Var}_{(\mathbb{P}_G)_{(\boldsymbol{s},h)}} \left[ V^*_{\hat{\mathcal{M}}_G} \right]}.$$

Substituting Eq. (29) into Eq. (28), we have

$$\left| (\hat{\mathbb{P}}_G - \mathbb{P}_G)_{(\boldsymbol{s},h)} V^*_{\hat{\mathcal{M}}_G} \right| \leq \left( \sqrt{\frac{4 \log 4|\mathcal{U}_{\boldsymbol{s}}||S||g|/\delta}{K'}} + 1 \right) \left\| V^*_{\hat{\mathcal{M}}_{G,s,u}} - V^*_{\hat{\mathcal{M}}_G} \right\|_\infty$$

$$+ \sqrt{\frac{4 \log 4|\mathcal{U}_{\boldsymbol{s}}||S||g|/\delta}{K'}} \sqrt{\mathrm{Var}_{(\mathbb{P}_G)_{(\boldsymbol{s},h)}} \left[ V^*_{\hat{\mathcal{M}}_G} \right]} + \frac{4 \log 4|\mathcal{U}_{\boldsymbol{s}}||S||g|/\delta}{3K'(1-\gamma)}. \tag{30}$$

By Lemmas 10 and 11, we have

$$\left\| V^*_{\hat{\mathcal{M}}_{G,s,u}} - V^*_{\hat{\mathcal{M}}_G} \right\|_\infty = \left\| V^*_{\hat{\mathcal{M}}_{G,s,u}} - V^*_{\hat{\mathcal{M}}_{G,s,V^*_{\hat{\mathcal{M}}_G}(\boldsymbol{s})}} \right\|_\infty \leq \left\| u - V^*_{\hat{\mathcal{M}}_G}(\boldsymbol{s}) \right\|_\infty, \tag{31}$$

where the first equality is because Lemma 10 and the second inequality is because Lemma 11. Substituting Eq. (31) into Eq. (30), we have

$$\left| (\hat{\mathbb{P}}_G - \mathbb{P}_G)_{(\boldsymbol{s},h)} V^*_{\hat{\mathcal{M}}_G} \right| \leq \left( \sqrt{\frac{4 \log 4|\mathcal{U}_{\boldsymbol{s}}||S||g|/\delta}{K'}} + 1 \right) \left\| V^*_{\hat{\mathcal{M}}_G} - u \right\|_\infty$$

$$+ \sqrt{\frac{4 \log 4|\mathcal{U}_{\boldsymbol{s}}||S||g|/\delta}{K'}} \sqrt{\mathrm{Var}_{(\mathbb{P}_G)_{(\boldsymbol{s},h)}} \left[ V^*_{\hat{\mathcal{M}}_G} \right]} + \frac{4 \log 4|\mathcal{U}_{\boldsymbol{s}}||S||g|/\delta}{3K'(1-\gamma)}. \tag{32}$$

We set $\mathcal{U}_{\boldsymbol{s}}$ as the uniformly spaced elements in $\left[V^*_{\mathcal{M}_G}(\boldsymbol{s}) - \frac{\gamma}{(1-\gamma)^2}\sqrt{\frac{2\log\left(\frac{4S|g|}{\delta}\right)}{K'}}, V^*_{\mathcal{M}_G}(\boldsymbol{s}) + \frac{\gamma}{(1-\gamma)^2}\sqrt{\frac{2\log\left(\frac{4S|g|}{\delta}\right)}{K'}}\right]$, and $|\mathcal{U}_{\boldsymbol{s}}| = \frac{1}{(1-\gamma)^2}$. We replace $\hat{\mathcal{M}}$ and $\mathcal{M}$ in Lemma 8 by $\hat{\mathcal{M}}_G$ and $\mathcal{M}_G$, respectively, and then we have

$$
\begin{aligned}
\min_{u \in \mathcal{U}} \left\| V^*_{\hat{\mathcal{M}}_G} - u \right\|_\infty &\leq (1-\gamma)^2 \frac{2\gamma}{(1-\gamma)^2} \sqrt{\frac{2\log\left(\frac{4S|g|}{\delta}\right)}{K'}} \\
&\leq 2\gamma \sqrt{\frac{2\log\left(\frac{4S|g|}{\delta}\right)}{K'}}.
\end{aligned}
$$

Plugging the above equation into Eq. (32), we have

$$
\begin{aligned}
\left|(\hat{\mathbb{P}}_G - \mathbb{P}_G)_{(\boldsymbol{s},h)} V^*_{\hat{\mathcal{M}}_G}\right| &\leq \frac{8\log\left(\frac{8S|g|}{\delta(1-\gamma)}\right)}{3K'(1-\gamma)} + 2\gamma\left(\sqrt{\frac{8\log\frac{8S|g|}{\delta(1-\gamma)}}{K'}} + 1\right)\sqrt{\frac{2\log\left(\frac{8S|g|}{\delta}\right)}{K'}} \\
&\quad + \sqrt{\frac{8\log\frac{8S|g|}{\delta(1-\gamma)}}{K'}}\sqrt{\mathrm{Var}_{(\mathbb{P}_G)_{(\boldsymbol{s},h)}}\left[V^*_{\hat{\mathcal{M}}_G}\right]} \\
&\leq \left(\frac{8}{3(1-\gamma)} + 8\gamma\right)\frac{\Delta_\delta}{K'} + 2\sqrt{2}\left(\gamma + \sqrt{\mathrm{Var}_{(\mathbb{P}_G)_{(\boldsymbol{s},h)}}\left[V^*_{\hat{\mathcal{M}}_G}\right]}\right)\sqrt{\frac{\Delta_\delta}{K'}},
\end{aligned}
\tag{33}
$$

where $\Delta_\delta := \log\left(\frac{8S|g|}{\delta(1-\gamma)}\right)$. Specifically, $\sqrt{\mathrm{Var}_{(\mathbb{P}_G)_{(\boldsymbol{s},h)}}\left[V^*_{\hat{\mathcal{M}}_G}\right]}$ can be rewritten as

$$
\begin{aligned}
\sqrt{\mathrm{Var}_{(\mathbb{P}_G)_{(\boldsymbol{s},h)}}\left[V^*_{\hat{\mathcal{M}}_G}\right]} &\leq \sqrt{\mathrm{Var}_{(\mathbb{P}_G)_{(\boldsymbol{s},h)}}\left[V^*_{\hat{\mathcal{M}}_G} - V^{\pi^\circ_T}_{\hat{\mathcal{M}}_G}\right]} + \sqrt{\mathrm{Var}_{(\mathbb{P}_G)_{(\boldsymbol{s},h)}}\left[V^{\pi^\circ_T}_{\hat{\mathcal{M}}_G} - V^{\pi^\circ_T}_{\mathcal{M}_G}\right]} \\
&\quad + \sqrt{\mathrm{Var}_{(\mathbb{P}_G)_{(\boldsymbol{s},h)}}\left[V^{\pi^\circ_T}_{\mathcal{M}_G}\right]}\left(\text{by } \sqrt{\mathrm{Var}_{\mathbb{P}}\left[X+Y\right]} \leq \sqrt{\mathrm{Var}_{\mathbb{P}}\left[X\right]} + \sqrt{\mathrm{Var}_{\mathbb{P}}\left[Y\right]}\right) \\
&\leq \left\|V^*_{\hat{\mathcal{M}}_G} - V^{\pi^\circ_T}_{\hat{\mathcal{M}}_G}\right\|_\infty + \left\|V^{\pi^\circ_T}_{\hat{\mathcal{M}}_G} - V^{\pi^\circ_T}_{\mathcal{M}_G}\right\|_\infty + \sqrt{\mathrm{Var}_{(\mathbb{P}_G)_{(\boldsymbol{s},h)}}\left[V^{\pi^\circ_T}_{\mathcal{M}_G}\right]} \\
&\leq \epsilon_{opt} + \left\|Q^{\pi^\circ_T}_{\hat{\mathcal{M}}_G} - Q^{\pi^\circ_T}_{\mathcal{M}_G}\right\|_\infty + \sqrt{\mathrm{Var}_{(\mathbb{P}_G)_{(\boldsymbol{s},h)}}\left[V^{\pi^\circ_T}_{\mathcal{M}_G}\right]} \\
&\quad \left(\text{by } \left\|V^*_{\hat{\mathcal{M}}_G} - V^{\pi^\circ_T}_{\hat{\mathcal{M}}_G}\right\|_\infty \leq \epsilon_{opt}\right).
\end{aligned}
\tag{34}
$$

We come back to the upper bound of $\left\|Q^{\pi^\circ_T}_{\hat{\mathcal{M}}_G} - Q^{\pi^\circ_T}_{\mathcal{M}_G}\right\|_\infty$.

$$
\begin{aligned}
\left\|Q^{\pi^\circ_T}_{\hat{\mathcal{M}}_G} - Q^{\pi^\circ_T}_{\mathcal{M}_G}\right\|_\infty &\leq \left\|\gamma(\mathbf{I} - \gamma\mathbb{P}^{\pi^\circ_T}_G)^{-1}(\hat{\mathbb{P}}_G - \mathbb{P}_G)V^*_{\hat{\mathcal{M}}_G}\right\|_\infty + \frac{\gamma\epsilon_{opt}}{1-\gamma} \quad \text{(by Eq. (27))} \\
&\leq \frac{\gamma}{1-\gamma}\left(\left(\frac{8}{3(1-\gamma)} + 8\gamma\right)\frac{\Delta_\delta}{K'} + 2\sqrt{2}\gamma\sqrt{\frac{\Delta_\delta}{K'}}\right) + \frac{\gamma\epsilon_{opt}}{1-\gamma} \\
&\quad + 2\sqrt{2}\gamma\left\|(\mathbf{I} - \gamma\mathbb{P}^{\pi^\circ_T}_G)^{-1}\sqrt{\mathrm{Var}_{\mathbb{P}_G}\left[V^*_{\hat{\mathcal{M}}_G}\right]}\sqrt{\frac{\Delta_\delta}{K'}}\right\|_\infty \quad \text{(by Eq. (33))} \\
&\leq \frac{\gamma}{1-\gamma}\left(\left(\frac{8}{3(1-\gamma)} + 8\gamma\right)\frac{\Delta_\delta}{K'} + 2\sqrt{2}\gamma\sqrt{\frac{\Delta_\delta}{K'}}\right) + \frac{\gamma\epsilon_{opt}}{1-\gamma}\left(1 + 2\sqrt{\frac{2\Delta_\delta}{K'}}\right) \\
&\quad + 2\gamma\sqrt{\frac{2\Delta_\delta}{K'}}\left\|(\mathbf{I} - \gamma\mathbb{P}^{\pi^\circ_T}_G)^{-1}\sqrt{\mathrm{Var}_{\mathbb{P}_G}\left[V^{\pi^\circ_T}_{\mathcal{M}_G}\right]}\right\|_\infty \\
&\quad + \frac{2\gamma}{1-\gamma}\sqrt{\frac{2\Delta_\delta}{K'}}\left\|Q^{\pi^\circ_T}_{\hat{\mathcal{M}}_G} - Q^{\pi^\circ_T}_{\mathcal{M}_G}\right\|_\infty \quad \text{(by Eq. (34))}.
\end{aligned}
$$

We replace $\mathbb{P}$ and $\pi$ in Lemma 9 by $\mathbb{P}_G$ and $\pi_T^\circ$, respectively, and then we have

$$\left\| (\mathbf{I} - \gamma \mathbb{P}_G^{\pi_T^\circ})^{-1} \sqrt{\mathrm{Var}_{\mathbb{P}_G}\left[ V_{\mathcal{M}_G}^{\pi_T^\circ} \right]} \right\|_\infty \leq \sqrt{\frac{2}{(1-\gamma)^3}}.$$

Plugging the above equation into $\left\| Q_{\hat{\mathcal{M}}_G}^{\pi_T^\circ} - Q_{\mathcal{M}_G}^{\pi_T^\circ} \right\|_\infty$, we then get

$$\left\| Q_{\hat{\mathcal{M}}_G}^{\pi_T^\circ} - Q_{\mathcal{M}_G}^{\pi_T^\circ} \right\|_\infty \leq \frac{\gamma}{1-\gamma} \left( \left( \frac{8}{3(1-\gamma)} + 8\gamma \right) \frac{\Delta_\delta}{K'} + 2\sqrt{2}\gamma \sqrt{\frac{\Delta_\delta}{K'}} \right) + \frac{\gamma \epsilon_{opt}}{1-\gamma} \left( 1 + 2\sqrt{\frac{2\Delta_\delta}{K'}} \right)$$
$$+ 4\gamma \sqrt{\frac{\Delta_\delta}{K'}} \sqrt{\frac{1}{(1-\gamma)^3}} + \frac{2\gamma}{1-\gamma} \sqrt{\frac{2\Delta_\delta}{K'}} \left\| Q_{\hat{\mathcal{M}}_G}^{\pi_T^\circ} - Q_{\mathcal{M}_G}^{\pi_T^\circ} \right\|_\infty.$$

Rearranging the above equation, we have

$$\left\| Q_{\hat{\mathcal{M}}_G}^{\pi_T^\circ} - Q_{\mathcal{M}_G}^{\pi_T^\circ} \right\|_\infty \leq \underbrace{\frac{1}{1 - \frac{2\gamma}{1-\gamma}\sqrt{\frac{2\Delta_\delta}{K'}}}}_{\mathsf{a}} \left( \underbrace{\frac{\gamma}{1-\gamma}\left( \frac{8}{3(1-\gamma)} + 8\gamma \right) \frac{\Delta_\delta}{K'}}_{\mathsf{b}} \right.$$
$$\left. + \underbrace{\left( \frac{2\sqrt{2}\gamma^2}{1-\gamma} + 4\gamma\sqrt{\frac{1}{(1-\gamma)^3}} \right) \sqrt{\frac{\Delta_\delta}{K'}}}_{\mathsf{c}} + \frac{\gamma\epsilon_{opt}}{1-\gamma}\left( 1 + 2\underbrace{\sqrt{\frac{2\Delta_\delta}{K'}}}_{\mathsf{d}} \right) \right).$$

When $K' \geq \frac{648\Delta_\delta}{(1-\gamma)^2}$, we have

$$\mathsf{a} \leq \frac{9}{8}, \ \mathsf{b} \leq \frac{8}{9}\gamma\sqrt{\frac{\Delta_\delta}{(1-\gamma)^3 K'}}, \ \mathsf{c} \leq 8\gamma\sqrt{\frac{\Delta_\delta}{(1-\gamma)^3 K'}}, \ \mathsf{d} \leq \frac{5}{6}.$$

Therefore, we get the upper bound of $\left\| Q_{\hat{\mathcal{M}}_G}^{\pi_T^\circ} - Q_{\mathcal{M}_G}^{\pi_T^\circ} \right\|_\infty$ as

$$\left\| Q_{\hat{\mathcal{M}}_G}^{\pi_T^\circ} - Q_{\mathcal{M}_G}^{\pi_T^\circ} \right\|_\infty \leq 10\gamma\sqrt{\frac{\Delta_\delta}{K'(1-\gamma)^3}} + \frac{3\gamma\epsilon_{opt}}{1-\gamma}.$$

With a similar derivation, we have

$$\left\| Q_{\mathcal{M}_G}^* - Q_{\hat{\mathcal{M}}_G}^{\pi^{\circ*}} \right\|_\infty \leq 10\gamma\sqrt{\frac{\Delta_\delta}{K'(1-\gamma)^3}}.$$

Plugging the above equations into Eq. (26), we finally get

$$\left\| V_{\mathcal{M}_G}^* - V_{\mathcal{M}_G}^{\pi_T^\circ} \right\|_\infty \leq 20\gamma\sqrt{\frac{\Delta_\delta}{K'(1-\gamma)^3}} + \frac{3\gamma\epsilon_{opt}}{1-\gamma} + \epsilon_{opt}$$
$$\leq 20\gamma\sqrt{\frac{S|g|\Delta_\delta}{K(1-\gamma)^3}} + \frac{4\epsilon_{opt}}{1-\gamma}.$$

We finally show $V_{\mathcal{M}_G}^* = V_{\mathcal{M}}^{\pi_G^*}$. Since $\pi_G^* \in \Pi_G$, there exists $\pi^{\circ*}$ such that $\pi_G^* = \pi^{\circ*} \circ \pi^{\mathsf{i}}$. For any $\pi^\circ : \mathcal{S} \to \Omega(\mathcal{G})$ is a higher-level policy, we can establish the equivalence between $V_{\mathcal{M}}^{\pi^\circ \circ \pi^{\mathsf{i}}}$ and $V_{\mathcal{M}_G}^{\pi^\circ}$

using the following reasoning.

$$V_{\mathcal{M}}^{\pi^\circ \circ \pi^i}(\boldsymbol{s}_0) = \sum_{h \in \mathcal{G}} \pi^\circ(h|\boldsymbol{s}_0) \mathbb{E}_{\boldsymbol{a}_0 \sim \pi^i(\cdot|\boldsymbol{s}_0, h)} \left[ R(\boldsymbol{s}_0, \boldsymbol{a}_0) + \gamma \sum_{\boldsymbol{s}_1} \mathbb{P}(\boldsymbol{s}_1|\boldsymbol{s}_0, \boldsymbol{a}_0) V_{\mathcal{M}}^{\pi^\circ \circ \pi^i}(\boldsymbol{s}_1) \right]$$

(by Bellman equation)

$$= \sum_{h \in \mathcal{G}} \pi^\circ(h|\boldsymbol{s}_0) \left( \mathbb{E}_{\boldsymbol{a}_0 \sim \pi^i(\cdot|\boldsymbol{s}_0, h)} \left[ R(\boldsymbol{s}_0, \boldsymbol{a}_0) \right] \right.$$

$$\left. + \gamma \sum_{\boldsymbol{s}_1} \mathbb{E}_{\boldsymbol{a}_0 \sim \pi^i(\cdot|\boldsymbol{s}_0, h)} \left[ \mathbb{P}(\boldsymbol{s}_1|\boldsymbol{s}_0, \boldsymbol{a}_0) \right] V_{\mathcal{M}}^{\pi^\circ \circ \pi^i}(\boldsymbol{s}_1) \right)$$

$$= \sum_{h \in \mathcal{G}} \pi^\circ(h|\boldsymbol{s}_0) \left( R_G(\boldsymbol{s}_0, h) + \gamma \sum_{\boldsymbol{s}_1} \mathbb{P}_G(\boldsymbol{s}_1|\boldsymbol{s}_0, h) V_{\mathcal{M}}^{\pi^\circ \circ \pi^i}(\boldsymbol{s}_1) \right)$$

$$= \mathbb{E}_{\tau_{\mathcal{M}_G}^{\pi^\circ}} \left[ \sum_{t=0}^{\infty} \gamma^t r_t | \boldsymbol{s}_0 \right] \quad \text{(by repeating above steps for infinite times)}$$

$$= V_{\mathcal{M}_G}^{\pi^\circ}(\boldsymbol{s}_0) \quad \text{(by definition of } V_{\mathcal{M}_G}).$$

(35)

Therefore,

$$\left\| V_{\mathcal{M}}^{\pi_G^*} - V_{\mathcal{M}}^{\pi_{G,T}} \right\|_\infty = \left\| V_{\mathcal{M}_G}^* - V_{\mathcal{M}_G}^{\pi_T^\circ} \right\|_\infty \le 20\gamma \sqrt{\frac{S|g|\Delta_\delta}{K(1-\gamma)^3}} + \frac{4\gamma\epsilon_{opt}}{1-\gamma}.$$

This concludes the proof of Lemma 2. $\qquad\square$

## C.5 SUPPLEMENTAL LEMMAS OF THEOREM 1

### C.5.1 MINIMIZATION OF $\beta_P$ AND $\beta_R$

We solve the $\beta_P$ and $\beta_R$ minimization problem (20) and (21) in this section.

We rewrite (20) as follows.

$$\begin{aligned}
\min \quad & \beta_P \\
\text{s.t.} \quad & \text{Eq. (1),} & (\boldsymbol{s}', \boldsymbol{s}, \boldsymbol{a}) \in \boldsymbol{\mathcal{S}} \times \boldsymbol{\mathcal{S}} \times \boldsymbol{\mathcal{A}}, \\
& \sum_{\boldsymbol{s}' \in \mathcal{S}} \mathbb{P}_1(\boldsymbol{s}'|\boldsymbol{s}, h) = 1, & (\boldsymbol{s}, h) \in \boldsymbol{\mathcal{S}} \times \mathcal{G}, \\
& \sum_{\boldsymbol{s}' \in \mathcal{S}} \mathbb{P}_2(\boldsymbol{s}'|\boldsymbol{s}, \boldsymbol{a}) = 1, & (\boldsymbol{s}, \boldsymbol{a}) \in \boldsymbol{\mathcal{S}} \times \boldsymbol{\mathcal{A}}, \\
& \mathbb{P}_1(\boldsymbol{s}'|\boldsymbol{s}, h) \ge 0, & (\boldsymbol{s}', \boldsymbol{s}, h) \in \boldsymbol{\mathcal{S}} \times \boldsymbol{\mathcal{S}} \times \mathcal{G}, \\
& \mathbb{P}_2(\boldsymbol{s}'|\boldsymbol{s}, \boldsymbol{a}) \ge 0, & (\boldsymbol{s}', \boldsymbol{s}, \boldsymbol{a}) \in \boldsymbol{\mathcal{S}} \times \boldsymbol{\mathcal{S}} \times \boldsymbol{\mathcal{A}}, \\
& 0 \le \boldsymbol{\beta}_P(\boldsymbol{s}, \boldsymbol{a}) \le \beta_P, & (\boldsymbol{s}, \boldsymbol{a}) \in \boldsymbol{\mathcal{S}} \times \boldsymbol{\mathcal{A}},
\end{aligned}$$

As the optimizations over $\beta_P(\boldsymbol{s}, h), \boldsymbol{s} \in \boldsymbol{\mathcal{S}}, h \in \mathcal{G}$ are independent, we can decompose the above optimization problem into $S|g|$ sub-problems. Each sub-problem seeks to find the minimum value of $\beta_P^*(\boldsymbol{s}, h)$ that satisfies the given constraints related to state $\boldsymbol{s}$ and group $h$. Consequently, $\beta_P^*$ is determined as the maximum of $\beta_P^*(\boldsymbol{s}, h)$ over all $\boldsymbol{s} \in \boldsymbol{\mathcal{S}}$ and $h \in \mathcal{G}$. We formally express this relationship as follows.

$$\beta_P^* = \max_{\boldsymbol{s} \in \boldsymbol{\mathcal{S}}, h \in \mathcal{G}} \beta_P^*(\boldsymbol{s}, h),$$

(36)

where $\beta_P^*(\boldsymbol{s}, h)$ is the solution to the following problem.

$$\begin{aligned}
\min \quad & \beta_P \\
\text{s.t.} \quad & \text{Eq. (1),} & (\boldsymbol{s}', \boldsymbol{a}) \in \boldsymbol{\mathcal{S}} \times \mathcal{A}_h, \\
& \sum_{\boldsymbol{s}' \in \mathcal{S}} \mathbb{P}_1(\boldsymbol{s}'|\boldsymbol{s}, h) = 1, \\
& \sum_{\boldsymbol{s}' \in \mathcal{S}} \mathbb{P}_2(\boldsymbol{s}'|\boldsymbol{s}, \boldsymbol{a}) = 1, & \boldsymbol{a} \in \mathcal{A}_h, \\
& \mathbb{P}_1(\boldsymbol{s}'|\boldsymbol{s}, h) \ge 0, & \boldsymbol{s}' \in \boldsymbol{\mathcal{S}}, \\
& \mathbb{P}_2(\boldsymbol{s}'|\boldsymbol{s}, \boldsymbol{a}) \ge 0, & (\boldsymbol{s}', \boldsymbol{a}) \in \boldsymbol{\mathcal{S}} \times \mathcal{A}_h. \\
& 0 \le \boldsymbol{\beta}_P(\boldsymbol{s}, \boldsymbol{a}) \le \beta_P, & \boldsymbol{a} \in \mathcal{A}_h.
\end{aligned}$$

(37)

Without loss of generality, we assume $\beta_P(s, a_1) = \beta_P^*(s, h)$, where $a_1 \in \mathcal{A}_h$. This implies that for any other actions $a$ such that $g(a_1) = g(a)$, we have $\beta_P(s, a) \leq \beta_P(s, a_1)$.

We firstly show the optimal solution $\beta_P^*(s, h)$ to Problem (36) is also the optimal solution to the following problem which relaxes the constraint $0 \leq \boldsymbol{\beta}_P(s, a) \leq \beta_P$.

$$
\begin{aligned}
\min \quad & \beta_P \\
\text{s.t.} \quad & \mathbb{P}(s'|s, a) = (1 - \beta_P)\mathbb{P}_1(s'|s, h) + \beta_P\mathbb{P}_2(s'|s, a), \quad (s', a) \in \mathcal{S} \times \mathcal{A}_h, \\
& \sum_{s' \in \mathcal{S}} \mathbb{P}_1(s'|s, h) = 1, \\
& \sum_{s' \in \mathcal{S}} \mathbb{P}_2(s'|s, a) = 1, & a \in \mathcal{A}_h, \\
& \mathbb{P}_1(s'|s, h) \geq 0, & s' \in \mathcal{S}, \\
& \mathbb{P}_2(s'|s, a) \geq 0, & (s', a) \in \mathcal{S} \times \mathcal{A}_h.
\end{aligned}
\tag{38}
$$

We verify the equivalence between Problem (37) and Problem (38) through two steps. We first show that $\beta_P^*(s, h)$ is a feasible solution to Problem (38), then we show $\beta_P^*(s, h)$ is a optimal to Problem (38).

(Step 1) Suppose $\mathbb{P}_1^*(\cdot|s, h)$ and $\mathbb{P}_2^*(\cdot|s, a)$ are the common and individual transition probability distribution when Problem (37) attains optimal. We construct $\mathbb{P}_1(\cdot|s, h) = \mathbb{P}_1^*(\cdot|s, h)$, and $\mathbb{P}_2(\cdot|s, a) = \frac{\beta_P^*(s,h) - \beta_P^*(s,a)}{\beta_P^*(s,h)}\mathbb{P}_1^*(\cdot|s, h) + \frac{\beta_P^*(s,a)}{\beta_P^*(s,h)}\mathbb{P}_2^*(\cdot|s, a)$. Then we can easily verify $\mathbb{P}_1(\cdot|s, h)$ and $\mathbb{P}_2(\cdot|s, a)$ satisfy constraints of problem (38), respectively.

(1) For the first constraint, we have

$$
\begin{aligned}
\mathbb{P}(\cdot|s, a) =& (1 - \beta_P^*(s, a))\mathbb{P}_1^*(\cdot|s, h) + \beta_P^*(s, a)\mathbb{P}_2^*(\cdot|s, a) \\
& (\beta_P^*(s, h), \mathbb{P}_1^*(\cdot|s, h), \text{ and } \mathbb{P}_2^*(\cdot|s, a) \text{ satisfy the first constraint of Problem (37)}) \\
=& (1 - \beta_P^*(s, h))\mathbb{P}_1(\cdot|s, h) + \beta_P^*(s, h)\mathbb{P}_2(\cdot|s, a) \\
& (\text{substituting } \mathbb{P}_1(\cdot|s, h) \text{ and } \mathbb{P}_2(\cdot|s, a)).
\end{aligned}
$$

(2) Since $\sum_{s' \in \mathcal{S}} \mathbb{P}_1(s'|s, h) = \sum_{s' \in \mathcal{S}} \mathbb{P}_1^*(s'|s, h) = 1$, the second constraint is satisfied.

(3) $\sum_{s' \in \mathcal{S}} \mathbb{P}_2(s'|s, h) = \frac{\beta_P^*(s,h) - \beta_P^*(s,a)}{\beta_P^*(s,h)} + \frac{\beta_P^*(s,a)}{\beta_P^*(s,h)} = 1$, then the third constraint is satisfied.

(4) By the definition of $\mathbb{P}_1^*$ and $\mathbb{P}_2^*$, we have $\mathbb{P}_1(s'|s, h) = \mathbb{P}_1^*(s'|s, h) \geq 0$ and $\mathbb{P}_2^*(s'|s, a) \geq 0$ for all $s' \in \mathcal{S}$.

(5) Since $0 \leq \beta_P^*(s, a) \leq \beta_P^*(s, h) \leq 1$, then $\frac{\beta_P^*(s,h) - \beta_P^*(s,a)}{\beta_P^*(s,h)} \geq 0$, and $\frac{\beta_P^*(s,a)}{\beta_P^*(s,h)} \geq 0$. We have $\mathbb{P}_2(s'|s, a) = \frac{\beta_P^*(s,h) - \beta_P^*(s,a)}{\beta_P^*(s,h)}\mathbb{P}_1^*(s'|s, h) + \frac{\beta_P^*(s,a)}{\beta_P^*(s,h)}\mathbb{P}_2^*(s'|s, a) \geq 0$ for all $s' \in \mathcal{S}$.

(Step 2) We then show that $\beta_P^*(s, h)$ is the optimal solution to Problem (38). If there exists $(\beta_P(s, h), \mathbb{P}_1(\cdot|s, h), \mathbb{P}_2(\cdot|s, a))$ that satisfy all constraints of Problem (38) and $\beta_P(s, h) < \beta_P^*(s, h)$, we arrive at a contradiction. In this case, $\beta_P(s, h)$ is also a feasible solution to sub-Problem Eq. (37), and it contradicts the assumption that $\beta_P^*(s, h)$ is the optimal solution to Problem (37).

We will focus on solving Problem (38). By the first constraint of problem (38), for any $a, a' \in \mathcal{A}_h$, we have

$$
\mathbb{P}(s'|s, a) - \mathbb{P}(s'|s, a_{s,h,s'}^*) = \beta_P(s, h)\left(\mathbb{P}_2(s'|s, a_{s,h,s'}^*) - \mathbb{P}_2(s'|s, a)\right).
\tag{39}
$$

Summing both sides of Eq. (39) over $s'$ and using $\sum_{s' \in \mathcal{S}} \mathbb{P}_2(s'|s, a) = 1$, we have

$$
1 - \sum_{s' \in \mathcal{S}} \mathbb{P}(s'|s, a_{s,h,s'}^*) = \beta_P(s, h)\left(1 - \sum_{s' \in \mathcal{S}} \mathbb{P}_2(s'|s, a_{s,h,s'}^*)\right).
$$

Therefore $\beta_P(s, h)$ can be rewritten as

$$
\begin{aligned}
\beta_P(s, h) &= \frac{1 - \sum_{s' \in \mathcal{S}} \mathbb{P}(s'|s, a_{s,h,s'}^*)}{1 - \sum_{s' \in \mathcal{S}} \mathbb{P}_2(s'|s, a_{s,h,s'}^*)} \\
&\geq 1 - \sum_{s' \in \mathcal{S}} \mathbb{P}(s'|s, a_{s,h,s'}^*) \quad (\text{by } \mathbb{P}_2(s'|s, a_{s,h,s'}^*) \geq 0) \\
&= 1 - \sum_{s' \in \mathcal{S}} \min_{a \in \mathcal{A}_h} \mathbb{P}(s'|s, a) \quad (\text{by } a_{s,h,s'}^* = \arg\min_{a \in \mathcal{A}_h} \mathbb{P}(s'|s, a)),
\end{aligned}
$$

where the inequality is tight when $\mathbb{P}_2(\boldsymbol{s}'|\boldsymbol{s}, \boldsymbol{a}_{\boldsymbol{s},h,\boldsymbol{s}'}^*) = 0$ for all $\boldsymbol{s}', \boldsymbol{s} \in \boldsymbol{\mathcal{S}}$. Thus we have

$$\beta_P^*(\boldsymbol{s}, h) = 1 - \sum_{\boldsymbol{s}' \in \boldsymbol{\mathcal{S}}} \min_{\boldsymbol{a} \in \mathcal{A}_h} \mathbb{P}(\boldsymbol{s}'|\boldsymbol{s}, \boldsymbol{a}).$$

Plugging $\beta_P^*(\boldsymbol{s}, h)$ into Eq. (36), we have $\beta_P^* = \max_{\boldsymbol{s} \in \boldsymbol{\mathcal{S}}, h \in \mathcal{G}}(1 - \sum_{\boldsymbol{s}' \in \boldsymbol{\mathcal{S}}} \min_{\boldsymbol{a} \in \mathcal{A}_h} \mathbb{P}(\boldsymbol{s}'|\boldsymbol{s}, \boldsymbol{a}))$.

$\beta_R$ minimization problem can be built in a similar way as the $\beta_P$ minimization problem.

$$
\begin{aligned}
\min \quad & \beta_R \\
\text{s.t.} \quad & \text{Eq. (2)}, \\
& 0 \le R_1(\boldsymbol{s}, h) \le 1, \quad (\boldsymbol{s}, h) \in \boldsymbol{\mathcal{S}} \times \mathcal{G}, \\
& 0 \le R_2(\boldsymbol{s}, \boldsymbol{a}) \le 1, \quad (\boldsymbol{s}, \boldsymbol{a}) \in \boldsymbol{\mathcal{S}} \times \boldsymbol{\mathcal{A}}, \\
& 0 \le \beta_R(\boldsymbol{s}, \boldsymbol{a}) \le \beta_R, \quad (\boldsymbol{s}, \boldsymbol{a}) \in \boldsymbol{\mathcal{S}} \times \boldsymbol{\mathcal{A}}.
\end{aligned}
\tag{40}
$$

Same to the discussion in the $\beta_P$ minimization, the optimization solution to the above problem is

$$\beta_R^* = \max_{\boldsymbol{s} \in \boldsymbol{\mathcal{S}}, h \in \mathcal{G}} \beta_R^*(\boldsymbol{s}, h), \tag{41}$$

where $\beta_R^*(\boldsymbol{s}, h)$ is the optimal solution to the followin problem.

$$
\begin{aligned}
\min \quad & \beta_R \\
\text{s.t.} \quad & R(\boldsymbol{s}, \boldsymbol{a}) = (1 - \beta_R)R_1(\boldsymbol{s}, h) + \beta_R R_2(\boldsymbol{s}, \boldsymbol{a}), \quad \boldsymbol{a} \in \mathcal{A}_h, \\
& 0 \le R_1(\boldsymbol{s}, h) \le 1, \\
& 0 \le R_2(\boldsymbol{s}, \boldsymbol{a}) \le 1, \quad \boldsymbol{a} \in \mathcal{A}_h.
\end{aligned}
\tag{42}
$$

By the first constraint in the above problem, for any $\boldsymbol{s} \in \boldsymbol{\mathcal{S}}, \boldsymbol{a}, \boldsymbol{a}' \in \mathcal{A}_h$ such that $R(\boldsymbol{s}, \boldsymbol{a}) \ne R(\boldsymbol{s}, \boldsymbol{a}')$, we have

$$
\begin{aligned}
\beta_R(\boldsymbol{s}, h) &= \frac{R(\boldsymbol{s}, \boldsymbol{a}) - R(\boldsymbol{s}, \boldsymbol{a}')}{R_2(\boldsymbol{s}, \boldsymbol{a}) - R_2(\boldsymbol{s}, \boldsymbol{a}')} \\
&\ge R(\boldsymbol{s}, \boldsymbol{a}) - R(\boldsymbol{s}, \boldsymbol{a}') \text{ (using } 0 \le R_2(\boldsymbol{s}, \boldsymbol{a}) \le 1).
\end{aligned}
$$

Therefore, $\beta_R(\boldsymbol{s}, h)$ should satisfy

$$
\begin{aligned}
\beta_R(\boldsymbol{s}, h) &\ge \max_{\boldsymbol{a}, \boldsymbol{a}' \in \mathcal{A}_h} (R(\boldsymbol{s}, \boldsymbol{a}) - R(\boldsymbol{s}, \boldsymbol{a}')) \\
&= \max_{\boldsymbol{a} \in \mathcal{A}_h} R(\boldsymbol{s}, \boldsymbol{a}) - \min_{\boldsymbol{a} \in \mathcal{A}_h} R(\boldsymbol{s}, \boldsymbol{a}),
\end{aligned}
$$

where inequality is tight when $R_2(\boldsymbol{s}, \boldsymbol{a}) = 1$ and $R_2(\boldsymbol{s}, \boldsymbol{a}') = 0$ with $\boldsymbol{a} = \arg\max_{\boldsymbol{a} \in \mathcal{A}_h} R(\boldsymbol{s}, \boldsymbol{a})$ and $\boldsymbol{a}' = \arg\min_{\boldsymbol{a} \in \mathcal{A}_h} R(\boldsymbol{s}, \boldsymbol{a})$. Thus $\beta_R^*(\boldsymbol{s}, h) = \max_{\boldsymbol{a} \in \mathcal{A}_h} R(\boldsymbol{s}, \boldsymbol{a}) - \min_{\boldsymbol{a} \in \mathcal{A}_h} R(\boldsymbol{s}, \boldsymbol{a})$. Plugging $\beta_R^*(\boldsymbol{s}, h)$ into Eq. (41), we have

$$\beta_R^*(g) = \max_{\boldsymbol{s} \in \boldsymbol{\mathcal{S}}, h \in \mathcal{G}} \left( \max_{\boldsymbol{a} \in \mathcal{A}_h} R(\boldsymbol{s}, \boldsymbol{a}) - \min_{\boldsymbol{a} \in \mathcal{A}_h} R(\boldsymbol{s}, \boldsymbol{a}) \right).$$

concludes the proof.

### C.5.2 SUPPLEMENTAL LEMMAS FOR ESTIMATION ERROR

**Lemma 3.** *For any MDP $\mathcal{M} = (\boldsymbol{\mathcal{S}}, \boldsymbol{\mathcal{A}}, \mathbb{P}, R, \gamma)$, the action value function with any policy $\pi : \boldsymbol{\mathcal{S}} \to \Omega(\boldsymbol{\mathcal{A}})$ can be written as*

$$Q_{\mathcal{M}}^\pi = (\mathbf{I} - \gamma \mathbb{P}^\pi)^{-1} R.$$

*Proof.* By definition of $Q_{\mathcal{M}}^\pi(\boldsymbol{s}, \boldsymbol{a})$, we have

$$
\begin{aligned}
Q_{\mathcal{M}}^\pi(\boldsymbol{s}, \boldsymbol{a}) &= \mathbb{E}_{\tau_{\mathcal{M}}^\pi} \left[ \sum_{t=0}^{\infty} \gamma^t r_t \mid s_0 = \boldsymbol{s}, a_0 = \boldsymbol{a} \right] \\
&= \sum_{t=0}^{\infty} \gamma^t \mathbb{P}^\pi(\boldsymbol{s}_t = \boldsymbol{s}', \boldsymbol{a}_t = \boldsymbol{a}'|s_0 = \boldsymbol{s}, a_0 = \boldsymbol{a}) R(\boldsymbol{s}', \boldsymbol{a}') \\
&= \sum_{t=0}^{\infty} (\gamma \mathbb{P}^\pi)^t R = (\mathbf{I} - \gamma \mathbb{P}^\pi)^{-1} R.
\end{aligned}
$$

$\square$

**Lemma 4.** *Let* $\left|V^*_{\hat{\mathcal{M}}_G} - V^{\pi^\circ_T}_{\hat{\mathcal{M}}_G}\right|_\infty \leq \epsilon_{opt}$. *Then*

$$\left\|Q^{\pi^{\circ*}}_{\hat{\mathcal{M}}_G} - Q^*_{\mathcal{M}_G}\right\|_\infty = \gamma \left\|(\mathbf{I} - \gamma\mathbb{P}^{\pi^{\circ*}}_G)^{-1}(\hat{\mathbb{P}}_G - \mathbb{P}_G)V^{\pi^{\circ*}}_{\hat{\mathcal{M}}_G}\right\|_\infty,$$

$$\left\|Q^{\pi^\circ_T}_{\hat{\mathcal{M}}_G} - Q^{\pi^\circ_T}_{\mathcal{M}_G}\right\|_\infty \leq \gamma \left\|(\mathbf{I} - \gamma\mathbb{P}^{\pi^\circ_T}_G)^{-1}(\hat{\mathbb{P}}_G - \mathbb{P}_G)V^*_{\hat{\mathcal{M}}_G}\right\|_\infty + \frac{\gamma\epsilon_{opt}}{1-\gamma}.$$

*Proof.* We have

$$
\begin{aligned}
\left\|Q^{\pi^{\circ*}}_{\hat{\mathcal{M}}_G} - Q^*_{\mathcal{M}_G}\right\|_\infty &= \left\|(\mathbf{I} - \gamma\hat{\mathbb{P}}^{\pi^{\circ*}}_G)^{-1}R - (\mathbf{I} - \gamma\mathbb{P}^{\pi^{\circ*}}_G)^{-1}R\right\|_\infty \quad \text{(by Lemma 3)} \\
&= \left\|(\mathbf{I} - \gamma\mathbb{P}^{\pi^{\circ*}}_G)^{-1}((\mathbf{I} - \gamma\mathbb{P}^{\pi^{\circ*}}_G) - (\mathbf{I} - \gamma\hat{\mathbb{P}}^{\pi^{\circ*}}_G))Q^{\pi^{\circ*}}_{\hat{\mathcal{M}}_G}\right\|_\infty \\
&= \gamma \left\|(\mathbf{I} - \gamma\mathbb{P}^{\pi^{\circ*}}_G)^{-1}(\hat{\mathbb{P}}^{\pi^{\circ*}}_G - \mathbb{P}^{\pi^{\circ*}}_G)Q^{\pi^{\circ*}}_{\hat{\mathcal{M}}_G}\right\|_\infty \\
&= \gamma \left\|(\mathbf{I} - \gamma\mathbb{P}^{\pi^{\circ*}}_G)^{-1}(\hat{\mathbb{P}}_G - \mathbb{P}_G)V^{\pi^{\circ*}}_{\hat{\mathcal{M}}_G}\right\|_\infty.
\end{aligned}
$$

Similar to the above derivation, we have

$$
\begin{aligned}
\left\|Q^{\pi^\circ_T}_{\hat{\mathcal{M}}_G} - Q^{\pi^\circ_T}_{\mathcal{M}_G}\right\|_\infty &= \gamma \left\|(\mathbf{I} - \gamma\mathbb{P}^{\pi^\circ_T}_G)^{-1}(\hat{\mathbb{P}}_G - \mathbb{P}_G)V^{\pi^\circ_T}_{\hat{\mathcal{M}}_G}\right\|_\infty \\
&= \gamma \left\|(\mathbf{I} - \gamma\mathbb{P}^{\pi^\circ_T}_G)^{-1}(\hat{\mathbb{P}}_G - \mathbb{P}_G)(V^{\pi^\circ_T}_{\hat{\mathcal{M}}_G} - V^*_{\hat{\mathcal{M}}_G} + V^*_{\hat{\mathcal{M}}_G})\right\|_\infty \\
&\leq \gamma \left\|(\mathbf{I} - \gamma\mathbb{P}^{\pi^\circ_T}_G)^{-1}(\hat{\mathbb{P}}_G - \mathbb{P}_G)(V^{\pi^\circ_T}_{\hat{\mathcal{M}}_G} - V^*_{\hat{\mathcal{M}}_G})\right\|_\infty \\
&\quad + \gamma \left\|(\mathbf{I} - \gamma\mathbb{P}^{\pi^\circ_T}_G)^{-1}(\hat{\mathbb{P}}_G - \mathbb{P}_G)V^*_{\hat{\mathcal{M}}_G})\right\|_\infty \\
&\leq \frac{\gamma\epsilon_{opt}}{1-\gamma} + \gamma \left\|(\mathbf{I} - \gamma\mathbb{P}^{\pi^\circ_T}_G)^{-1}(\hat{\mathbb{P}}_G - \mathbb{P}_G)V^*_{\hat{\mathcal{M}}_G})\right\|_\infty \\
&\quad \text{(by } \left\|V^{\pi^\circ_T}_{\hat{\mathcal{M}}_G} - V^*_{\hat{\mathcal{M}}_G}\right\|_\infty \leq \epsilon_{opt}).
\end{aligned}
$$

$\square$

**Lemma 5** (Hoeffding's inequality for general bounded random variables (Vershynin (2018), Theorem 2.2.6)). *Let* $X_1, \cdots, X_N$ *be independent random variables. Assume that* $X_i \in [m_i, M_i]$ *for every* $i$. *Then, for any* $t > 0$, *we have*

$$\mathbb{P}\left\{\sum_{i=1}^N (X_i - \mathbb{E}X_i) \geq t\right\} \leq \exp\left(-\frac{2t^2}{\sum_{i=1}^N (M_i - m_i)^2}\right).$$

**Lemma 6** (Bernstein's inequality for bounded distributions (Vershynin (2018), Theorem 2.8.4)). *Let* $X_1, \cdots, X_N$ *be independent, mean zero random variables, such that* $|X_i| \leq B$ *for all* $i$. *Then for every* $t \geq 0$, *we have*

$$\mathbb{P}\left\{\left|\sum_{i=1}^N X_i\right| \geq t\right\} \leq 2\exp\left(-\frac{t^2/2}{\sigma^2 + Bt/3}\right), \tag{43}$$

*where* $\sigma^2 = \sum_{i=1}^N \mathrm{Var}\left[X_i\right]$.

**Lemma 7** (Variant of Bernstein's inequality). *Let* $X_1, \cdots, X_N$ *be independent, identically distributed, mean zero random variables, such that* $|X_i| \leq B$ *and* $\mathrm{Var}\left[X_i\right] = \mathrm{Var}\left[X\right]$ *for all* $i$. *Then with probability exceeding* $1 - \delta/2$, *we have*

$$\frac{1}{N}\left|\sum_{i=1}^N X_i\right| \leq \frac{4B}{3N}\log\frac{4}{\delta} + \sqrt{\frac{4\mathrm{Var}\left[X\right]}{N}\log\frac{4}{\delta}}. \tag{44}$$

*Proof.* When $X_1, \cdots, X_N$ are identically distributed and $\mathrm{Var}\left[X_i\right] = \mathrm{Var}\left[X\right]$ for all $i$, Eq. (43) can be rewritten as

$$\mathbb{P}\left\{\frac{1}{N}\left|\sum_{i=1}^N X_i\right| \geq t\right\} \leq 2\exp\left(-\frac{Nt^2/2}{\mathrm{Var}\left[X\right] + Bt/3}\right).$$

Let

$$2 \exp\left(-\frac{Nt^2/2}{\text{Var}[X] + Bt/3}\right) \leq \frac{\delta}{2}. \tag{45}$$

we can rewrite the above equation as

$$t^2 \geq \frac{2\text{Var}[X]}{N} \log \frac{4}{\delta} + \frac{2Bt}{3N} \log \frac{4}{\delta}.$$

A sufficient condition of $t$ satisfying the above inequality is

$$t^2 \geq \frac{4\text{Var}[X]}{N} \log \frac{4}{\delta} \quad \text{and} \quad t^2 \geq \frac{4Bt}{3N} \log \frac{4}{\delta}.$$

Therefore, when $t \geq \frac{4B}{3N} \log \frac{4}{\delta} + \sqrt{\frac{4\text{Var}[X]}{N} \log \frac{4}{\delta}}$,

$$\mathbb{P}\left\{\frac{1}{N}\left|\sum_{i=1}^{N} X_i\right| \geq t\right\} \leq \frac{\delta}{2}.$$

On the other words, with probability exceeding $1 - \delta/2$, we have

$$\frac{1}{N}\left|\sum_{i=1}^{N} X_i\right| \leq \frac{4B}{3N} \log \frac{4}{\delta} + \sqrt{\frac{4\text{Var}[X]}{N} \log \frac{4}{\delta}}.$$

$\square$

**Lemma 8** (Gheshlaghi Azar et al. (2013), Lemma 4). *Consider MDP $\mathcal{M} = \{\mathcal{S}, \mathcal{A}, \mathbb{P}, R, \gamma\}$ which satisfies Assumption 1. $\hat{\mathcal{M}}$ is a estimation of $\mathcal{M}$ based on the generative model with $n$ samples for each state-action pair. With a probability exceeding $\delta$, one has*

$$\|V_{\mathcal{M}}^* - V_{\hat{\mathcal{M}}}^{\pi_{\hat{\mathcal{M}}}^*}\|_\infty \leq \frac{\gamma}{(1-\gamma)^2}\sqrt{\frac{2\log\left(\frac{2SA}{\delta}\right)}{n}},$$

$$\|V_{\mathcal{M}}^* - V_{\hat{\mathcal{M}}}^*\|_\infty \leq \frac{\gamma}{(1-\gamma)^2}\sqrt{\frac{2\log\left(\frac{2SA}{\delta}\right)}{n}}.$$

**Lemma 9** (Agarwal et al. (2020b) Lemma 5). *For any policy $\pi$ and MDP $\mathcal{M} = \{\mathcal{S}, \mathcal{A}, \mathbb{P}, R, \gamma\}$,*

$$\left\|(\mathbf{I} - \gamma\mathbb{P}^\pi)^{-1}\sqrt{\text{Var}_{\mathbb{P}}[V_{\mathcal{M}}^\pi]}\right\|_\infty \leq \sqrt{\frac{2}{(1-\gamma)^3}}.$$

**Lemma 10** (Agarwal et al. (2020b) Lemma 8). *Let $u^* = V_{\mathcal{M}}^*(s)$ and $u^\pi = V_{\mathcal{M}}^\pi(s)$. We have*

$$V_{\mathcal{M}}^* = V_{\mathcal{M},s,u^*}^*, \quad \text{and for all policies } \pi, \quad V_{\mathcal{M}}^\pi = V_{V_{\mathcal{M}},s,u^\pi}^\pi.$$

**Lemma 11** (Agarwal et al. (2020b) Lemma 9). *For all states $s, u, u' \in \mathbb{R}$, and policies $\pi$,*

$$\left\|Q_{\mathcal{M},s,u}^* - Q_{\mathcal{M},s,u'}^*\right\|_\infty \leq |u - u'| \quad \text{and} \quad \left\|Q_{\mathcal{M},s,u}^\pi - Q_{\mathcal{M},s,u'}^\pi\right\|_\infty \leq |u - u'|.$$

Lemma 11 implies

$$\left\|V_{\mathcal{M},s,u}^* - V_{\mathcal{M},s,u'}^*\right\|_\infty \leq \left\|Q_{\mathcal{M},s,u}^* - Q_{\mathcal{M},s,u'}^*\right\|_\infty \leq |u - u'|.$$

## D UPPER-BOUND OF PERFORMANCE LOSS WITH VALUE ITERATION

We consider value iteration (VI)—a specific dynamic programming algorithm—as shown in Algorithm 2. Algorithm 2 provides generative-model-based value iteration, where $\hat{Q}^{t+1} = \mathcal{T}_{\hat{\mathcal{M}}_G}\hat{Q}^t$ for each iteration $t$ and the output policy is $\pi_t^\circ(s) = \arg\max_h \hat{Q}^t(s, h)$.

**Corollary 1.** *Let that Algorithm 2 be the dynamic programming algorithm in Algorithm 1. When sample complexity and computational complexity are $\mathcal{C}_{samp}(K) = K$ and $\mathcal{C}_{comp}(|g|, T) = (S^2|g| + 2S|g|)T$, respectively, With a probability larger than $1 - \delta$,*

$$\left\| V_{\mathcal{M}}^* - V_{\mathcal{M}}^{\pi_{G,T}} \right\|_\infty \leq \epsilon_{perf}(g, K, T),$$

*where*

$$\epsilon_{perf}(g, K, T) = 2\left( \frac{\beta_R^*(g)}{1-\gamma} + \frac{\gamma\beta_P^*(g)}{(1-\gamma)^2} \right) + 20\gamma\sqrt{\frac{S|g|\log\left(\frac{8S|g|}{\delta(1-\gamma)}\right)}{K(1-\gamma)^3}} + \frac{8\gamma^T}{(1-\gamma)^3}.$$

---

**Algorithm 2** Value Iteration

---

1: **Input:** $\hat{\mathcal{M}}_G = \{\mathcal{S}, \mathcal{G}, A, \hat{\mathbb{P}}_G, \hat{R}_G, \gamma\}$.
2: **Output:** policy $\pi_T^\circ$.
3: **for** $t = 1, \cdots, T$ **do**
4:   **for** $(s, h) \in \mathcal{S} \times \mathcal{G}$ **do**
5:    $\hat{Q}^{t+1}(s, h) = \hat{R}_G(s, h) + \gamma\langle\hat{\mathbb{P}}_G(\cdot|s, h), \max_{h' \in \mathcal{G}} \hat{Q}^t(\cdot, h')\rangle$.
6:   **end for**
7: **end for**
8: Output policy $\pi_T^\circ(s) = \arg\max_{h \in \mathcal{G}} \hat{Q}^T(s, h), s \in \mathcal{S}$.

---

Previous research has demonstrated that the value function of the resulting policy converges to that of the optimal policy under $\hat{\mathcal{M}}_G$ at a linear rate Munos (2005); Munos and Szepesvári (2008). This convergence is formalized in the following lemma.

**Lemma 12.** *Let $\pi_T^\circ$ be the output policy of Algorithm 2 after $T$ iterations. Then we have*

$$\left\| V_{\hat{\mathcal{M}}_G}^* - V_{\hat{\mathcal{M}}_G}^{\pi_T^\circ} \right\|_\infty \leq \frac{2\gamma^T}{(1-\gamma)^2}.$$

For the completeness of this paper, we provide the proof of Lemma 12 in Appendix D.1.

Plugging Lemma 12 into Theorem 1 leads to Corollary 1.

## D.1 ALGORITHM ERROR

*Proof.* By Bellman optimality equation and VI update rule, we have $Q_{\hat{\mathcal{M}}_G}^*(s, h) = \mathcal{T}_{\hat{\mathcal{M}}_G} Q_{\hat{\mathcal{M}}_G}^*(s, h)$ and $\hat{Q}^T(s, h) = \mathcal{T}_{\hat{\mathcal{M}}_G} \hat{Q}^{T-1}(s, h)$, respectively, where $\mathcal{T}_{\hat{\mathcal{M}}_G}$ is the Bellman operator under $\hat{\mathcal{M}}_G$. We can write $\left| Q_{\hat{\mathcal{M}}_G}^*(s, h) - \hat{Q}^T(s, h) \right|$ as

$$
\begin{aligned}
\left| Q_{\hat{\mathcal{M}}_G}^*(s, h) - \hat{Q}^T(s, h) \right| &= \left| \mathcal{T}_{\hat{\mathcal{M}}_G} Q_{\hat{\mathcal{M}}_G}^*(s, h) - \mathcal{T}_{\hat{\mathcal{M}}_G} \hat{Q}^{T-1}(s, h) \right| \\
&= \gamma \left| \langle \mathbb{P}_G(\cdot|s, h), \max_{h'} Q_{\hat{\mathcal{M}}_G}^*(\cdot, h') - \max_{h'} \hat{Q}^{T-1}(\cdot, h') \rangle \right| \\
&\qquad\qquad\qquad\qquad\qquad \text{(by definition of } \mathcal{T}_{\mathcal{M}} f) \\
&= \gamma \left\langle \mathbb{P}_G(\cdot|s, h), \left| \max_{h'} Q_{\hat{\mathcal{M}}_G}^*(\cdot, h') - \max_{h''} \hat{Q}^{T-1}(\cdot, h'') \right| \right\rangle \\
&\qquad\qquad\qquad\qquad\qquad \text{(by triangle inequality)} \\
&= \gamma \left\langle \mathbb{P}_G(\cdot|s, h), \max_{h'} \left| Q_{\hat{\mathcal{M}}_G}^*(\cdot, h') - \hat{Q}^{T-1}(\cdot, h') \right| \right\rangle \\
&\leq \gamma \max_{s', h'} \left| Q_{\hat{\mathcal{M}}_G}^*(s', h') - \hat{Q}^{T-1}(s', h') \right| \quad \text{(by } \sum_{s'} \mathbb{P}_G(s'|s, h) = 1) \\
&\leq \gamma \left\| Q_{\hat{\mathcal{M}}_G}^* - \hat{Q}^{T-1} \right\|_\infty.
\end{aligned}
$$

Maximizing both sides of over $(\boldsymbol{s}, h)$, we have

$$\left\|Q^*_{\hat{\mathcal{M}}_G} - \hat{Q}^T\right\|_\infty \leq \gamma \left\|Q^*_{\hat{\mathcal{M}}_G} - \hat{Q}^{T-1}\right\|_\infty. \tag{46}$$

Since $\hat{Q}^t \leq \frac{1}{1-\gamma}\mathbf{1}$, we can apply the Eq. (46) over $\left\|Q^*_{\hat{\mathcal{M}}_G} - \hat{Q}^T\right\|_\infty$ for $T$ steps and get

$$
\begin{aligned}
\left\|Q^*_{\hat{\mathcal{M}}_G} - \hat{Q}^T\right\|_\infty &\leq \gamma^T \left\|Q^*_{\hat{\mathcal{M}}_G} - \hat{Q}^0\right\|_\infty \\
&= \frac{\gamma^T}{1-\gamma} \quad \text{(by } 0 \leq Q^{\pi^\circ}_{\mathcal{M}_G}, \hat{Q}^T \leq \frac{1}{1-\gamma}).
\end{aligned} \tag{47}
$$

Then we can write $V^*_{\hat{\mathcal{M}}_G}(\boldsymbol{s}) - V^{\pi^\circ_T}_{\hat{\mathcal{M}}_G}(\boldsymbol{s})$ as

$$
\begin{aligned}
V^*_{\hat{\mathcal{M}}_G}(\boldsymbol{s}) - V^{\pi^\circ_T}_{\hat{\mathcal{M}}_G}(\boldsymbol{s}) &= Q^*_{\hat{\mathcal{M}}_G}(\boldsymbol{s}, \pi^{\circ*}(\boldsymbol{s})) - Q^{\pi^\circ_T}_{\hat{\mathcal{M}}_G}(\boldsymbol{s}, \pi^\circ_T(\boldsymbol{s})) \\
&= Q^*_{\hat{\mathcal{M}}_G}(\boldsymbol{s}, \pi^{\circ*}(\boldsymbol{s})) + \left(-\hat{Q}^T(\boldsymbol{s}, \pi^{\circ*}(\boldsymbol{s})) + \hat{Q}^T(\boldsymbol{s}, \pi^{\circ*}(\boldsymbol{s}))\right) + \left(-\hat{Q}^T(\boldsymbol{s}, \pi^\circ_T(\boldsymbol{s})) + \hat{Q}^T(\boldsymbol{s}, \pi^\circ_T(\boldsymbol{s}))\right) \\
&\quad + \left(-Q^*_{\hat{\mathcal{M}}_G}(\boldsymbol{s}, \pi^\circ_T(\boldsymbol{s}) + Q^*_{\hat{\mathcal{M}}_G}(\boldsymbol{s}, \pi^\circ_T(\boldsymbol{s}))\right) - Q^{\pi^\circ_T}_{\hat{\mathcal{M}}_G}(\boldsymbol{s}, \pi^\circ_T(\boldsymbol{s})) \\
&\leq \left(Q^*_{\hat{\mathcal{M}}_G}(\boldsymbol{s}, \pi^{\circ*}(\boldsymbol{s})) - \hat{Q}^T(\boldsymbol{s}, \pi^{\circ*}(\boldsymbol{s}))\right) + \left(\hat{Q}^T(\boldsymbol{s}, \pi^\circ_T(\boldsymbol{s})) - Q^*_{\hat{\mathcal{M}}_G}(\boldsymbol{s}, \pi^\circ_T(\boldsymbol{s}))\right) \\
&\quad + \left(Q^*_{\hat{\mathcal{M}}_G}(\boldsymbol{s}, \pi^\circ_T(\boldsymbol{s})) - Q^{\pi^\circ_T}_{\hat{\mathcal{M}}_G}(\boldsymbol{s}, \pi^\circ_T(\boldsymbol{s}))\right),
\end{aligned}
$$

where the inequality is because $\hat{Q}^T(\boldsymbol{s}, \pi^{\circ*}(\boldsymbol{s}))) - \hat{Q}^T(\boldsymbol{s}, \pi^\circ_T) \leq 0$ since $\pi^\circ_T(\boldsymbol{s}) = \arg\max_h \hat{Q}^T(\boldsymbol{s}, h)$.

Since $V^*_{\hat{\mathcal{M}}_G}(\boldsymbol{s}) - V^{\pi^\circ_T}_{\hat{\mathcal{M}}_G}(\boldsymbol{s}) \geq 0$, we apply infinity norm on both sides of the above equation and get

$$
\begin{aligned}
&\left\|V^*_{\hat{\mathcal{M}}_G} - V^{\pi^\circ_T}_{\hat{\mathcal{M}}_G}\right\|_\infty \\
&\leq \max_{\boldsymbol{s}} \Big|(Q^*_{\hat{\mathcal{M}}_G}(\boldsymbol{s}, \pi^{\circ*}(\boldsymbol{s})) - \hat{Q}^T(\boldsymbol{s}, \pi^{\circ*}(\boldsymbol{s}))) + (\hat{Q}^T(\boldsymbol{s}, \pi^\circ_T(\boldsymbol{s})) - Q^*_{\hat{\mathcal{M}}_G}(\boldsymbol{s}, \pi^\circ_T(\boldsymbol{s}))) \\
&\qquad\qquad + (Q^*_{\hat{\mathcal{M}}_G}(\boldsymbol{s}, \pi^\circ_T(\boldsymbol{s})) - Q^{\pi^\circ_T}_{\hat{\mathcal{M}}_G}(\boldsymbol{s}, \pi^\circ_T(\boldsymbol{s})))\Big| \\
&\leq \underbrace{\max_{\boldsymbol{s}} \left|Q^*_{\hat{\mathcal{M}}_G}(\boldsymbol{s}, \pi^{\circ*}(\boldsymbol{s})) - \hat{Q}^T(\boldsymbol{s}, \pi^{\circ*}(\boldsymbol{s}))\right| + \max_{\boldsymbol{s}} \left|\hat{Q}^T(\boldsymbol{s}, \pi^\circ_T(\boldsymbol{s})) - Q^*_{\hat{\mathcal{M}}_G}(\boldsymbol{s}, \pi^\circ_T(\boldsymbol{s}))\right|}_{\mathtt{A}_1} \\
&\quad + \underbrace{\max_{\boldsymbol{s}} \left|Q^*_{\hat{\mathcal{M}}_G}(\boldsymbol{s}, \pi^\circ_T(\boldsymbol{s})) - Q^{\pi^\circ_T}_{\hat{\mathcal{M}}_G}(\boldsymbol{s}, \pi^\circ_T(\boldsymbol{s}))\right|}_{\mathtt{A}_2} \quad \text{(by triangle inequality)}.
\end{aligned}
$$

We bound $\mathtt{A}_1$ and $\mathtt{A}_2$ separately.

$$
\begin{aligned}
\mathtt{A}_1 &\leq \max_{\boldsymbol{s},h} \left|Q^*_{\hat{\mathcal{M}}_G}(\boldsymbol{s}, h) - \hat{Q}^T(\boldsymbol{s}, h)\right| + \max_{\boldsymbol{s},h} \left|\hat{Q}^T(\boldsymbol{s}, h) - Q^*_{\hat{\mathcal{M}}_G}(\boldsymbol{s}, h)\right| \\
&\leq 2 \left\|Q^*_{\hat{\mathcal{M}}_G} - \hat{Q}^T\right\|_\infty.
\end{aligned}
$$

Then,

$$
\begin{aligned}
\mathtt{A}_2 &\leq \max_{\boldsymbol{s}} \left|Q^*_{\hat{\mathcal{M}}_G}(\boldsymbol{s}, \pi^\circ_T(\boldsymbol{s})) - Q^{\pi^\circ_T}_{\hat{\mathcal{M}}_G}(\boldsymbol{s}, \pi^\circ_T(\boldsymbol{s}))\right| \\
&= \max_{\boldsymbol{s}} \Big|R(\boldsymbol{s}, \pi^\circ_T(\boldsymbol{s})) + \gamma\langle\hat{\mathbb{P}}_G(\cdot|\boldsymbol{s}, \pi^\circ_T(\boldsymbol{s})), V^*_{\hat{\mathcal{M}}_G}(\cdot)\rangle \\
&\qquad\qquad - (R(\boldsymbol{s}, \pi^\circ_T(\boldsymbol{s})) + \gamma\langle\hat{\mathbb{P}}_G(\cdot|\boldsymbol{s}, \pi^\circ_T(\boldsymbol{s}))), V^{\pi^\circ_T}_{\hat{\mathcal{M}}_G}(\cdot)\rangle)\Big| \quad \text{(by Bellman's Equation)} \\
&= \gamma \max_{\boldsymbol{s}} \left|\langle\hat{\mathbb{P}}_G(\cdot|\boldsymbol{s}, \pi^\circ_T(\boldsymbol{s})), V^*_{\hat{\mathcal{M}}_G}(\cdot) - V^{\pi^\circ_T}_{\hat{\mathcal{M}}_G}(\cdot)\rangle\right| \\
&\leq \gamma \max_{\boldsymbol{s}} \left\langle\hat{\mathbb{P}}_G(\cdot|\boldsymbol{s}, \pi^\circ_T(\boldsymbol{s})), \left|V^*_{\hat{\mathcal{M}}_G}(\cdot) - V^{\pi^\circ_T}_{\hat{\mathcal{M}}_G}(\cdot)\right|\right\rangle \leq \gamma \left\|V^*_{\hat{\mathcal{M}}_G} - V^{\pi^\circ_T}_{\hat{\mathcal{M}}_G}\right\|_\infty.
\end{aligned}
$$

Plugging the upper bound of $A_1$ and $A_2$ into $\left\| V^*_{\hat{\mathcal{M}}_G} - V^{\pi^\circ_T}_{\hat{\mathcal{M}}_G} \right\|_\infty$ and rearranging the equation, we have

$$\left\| V^*_{\hat{\mathcal{M}}_G} - V^{\pi^\circ_T}_{\hat{\mathcal{M}}_G} \right\|_\infty \leq \frac{2 \left\| Q^*_{\hat{\mathcal{M}}_G} - \hat{Q}^T \right\|_\infty}{1-\gamma} \leq \frac{2\gamma^T}{(1-\gamma)^2},$$

where the last inequality is because Eq. (47).

This concludes the proof of Lemma 12. $\qquad\square$

## E PROOF OF PROPOSITION 1

The approximate grouping function optimization problem can be rewritten as

$$\max_{g \in \mathcal{D}} f(\hat{\epsilon}_{\text{perf}}(g, K^*(|g|), T^*(|g|)), \mathcal{C}_{\text{samp}}(K^*(|g|)), \mathcal{C}_{\text{comp}}(|g|, T^*(|g|))), \tag{48}$$

where

$$\hat{\epsilon}_{\text{perf}}(g, K^*(|g|), T^*(|g|)) = 2\left( \frac{\gamma\hat{\beta}^*_P(|g|)}{(1-\gamma)^2} + \frac{\hat{\beta}^*_R(|g|)}{1-\gamma} \right) + 20\gamma\sqrt{\frac{S|g|\log\left(\frac{8S|g|}{\delta(1-\gamma)}\right)}{K(1-\gamma)^3}} + \frac{4\epsilon_{opt}}{1-\gamma},$$

$$\hat{\beta}^*_P(|g|) = \max_{s \in \mathcal{S}, h \in \mathcal{G}} \left( 1 - \sum_{s'} \min_{a \in \bar{\mathcal{A}}_h} \hat{\mathbb{P}}(s'|s, a) \right),$$

$$\hat{\beta}^*_R(|g|) = \max_{s \in \mathcal{S}, g(a_1) = g(a_2)} (R(s, a_1) - R(s, a_2)). \tag{49}$$

For notational simplicity, we write $f(\hat{\epsilon}_{\text{perf}}(g, K^*(|g|), T^*(|g|)), \mathcal{C}_{\text{samp}}(K^*(|g|)), \mathcal{C}_{\text{comp}}(|g|, T^*(|g|)))$, the optimization objective function in Eq. (48) that maps from $\mathcal{G}$ to $\mathcal{R}$, as $\hat{f} : \mathcal{G} \to \mathcal{R}$.

*Proof.* We first show for any MDP satisfying Eq. (7), the approximation error of $\beta^*_P(|g|)$ is bounded by terms related to $\eta_P$.

$$\beta^*_P(|g|) = \max_{s \in \mathcal{S}, h \in \mathcal{G}} \left( 1 - \sum_{s'} \min_{a \in \mathcal{A}_h} \mathbb{P}(s'|s, a) \right)$$

$$= \max_{s \in \mathcal{S}, h \in \mathcal{G}} \left( \max_{a \in \mathcal{A}_h} \sum_{s'} \mathbb{P}(s'|s, a) - \sum_{s'} \min_{a \in \mathcal{A}_h} \mathbb{P}(s'|s, a) \right) \quad \text{(by } \sum_{s'} \mathbb{P}(s'|s, a) \text{ for any } a)$$

$$\leq \max_{s \in \mathcal{S}, h \in \mathcal{G}} \left( \sum_{s'} \max_{a \in \mathcal{A}_h} \mathbb{P}(s'|s, a) - \sum_{s'} \min_{a \in \mathcal{A}_h} \mathbb{P}(s'|s, a) \right)$$

$$= \max_{s \in \mathcal{S}, h \in \mathcal{G}} \left( \sum_{s'} \left( \max_{a \in \mathcal{A}_h} \mathbb{P}(s'|s, a) - \min_{a \in \mathcal{A}_h} \mathbb{P}(s'|s, a) \right) \right)$$

$$= \max_{s \in \mathcal{S}, h \in \mathcal{G}} \left( \sum_{s'} \max_{a_1, a_2 \in \mathcal{A}_h} (\mathbb{P}(s'|s, a_1) - \mathbb{P}(s'|s, a_2)) \right)$$

$$\leq \sum_{s'} \left( \max_{s \in \mathcal{S}, h \in \mathcal{G}} \max_{a_1, a_2 \in \mathcal{A}_h} (\mathbb{P}(s'|s, a_1) - \mathbb{P}(s'|s, a_2)) \right)$$

$$\leq S\eta_P \qquad \text{(by Eq. (7))}.$$

Since $0 < \beta^*_P(|g|), \hat{\beta}^*_P(|g|) < 1$, we can write $\beta^*_P(|g|) - \hat{\beta}^*_P(|g|)$ as

$$\beta^*_P(|g|) - \hat{\beta}^*_P(|g|) \leq \beta^*_P(|g|) \leq S\eta_P.$$

Define $\boldsymbol{a}^*_{\boldsymbol{s},h} = \arg\min_{\boldsymbol{a}\in\bar{\mathcal{A}}_h} \hat{\mathbb{P}}(\boldsymbol{s}'|\boldsymbol{s},\boldsymbol{a})$. We can rewrite $\hat{\beta}^*_P(|g|) - \beta^*_P(|g|)$ as

$$
\begin{aligned}
\hat{\beta}^*_P(|g|) - \beta^*_P(|g|) &= \max_{\boldsymbol{s}\in\boldsymbol{\mathcal{S}},h\in\mathcal{G}} \left(1 - \sum_{\boldsymbol{s}'} \min_{\boldsymbol{a}\in\bar{\mathcal{A}}_h} \hat{\mathbb{P}}(\boldsymbol{s}'|\boldsymbol{s},\boldsymbol{a})\right) - \max_{\boldsymbol{s}\in\boldsymbol{\mathcal{S}},h\in\mathcal{G}} \left(1 - \sum_{\boldsymbol{s}'} \min_{\boldsymbol{a}\in\mathcal{A}_h} \mathbb{P}(\boldsymbol{s}'|\boldsymbol{s},\boldsymbol{a})\right) \\
&= \max_{\boldsymbol{s}\in\boldsymbol{\mathcal{S}},h\in\mathcal{G}} \left(\sum_{\boldsymbol{s}'} \min_{\boldsymbol{a}\in\mathcal{A}_h} \mathbb{P}(\boldsymbol{s}'|\boldsymbol{s},\boldsymbol{a}) - \sum_{\boldsymbol{s}'} \min_{\boldsymbol{a}\in\bar{\mathcal{A}}_h} \hat{\mathbb{P}}(\boldsymbol{s}'|\boldsymbol{s},\boldsymbol{a})\right) \\
&\leq \max_{\boldsymbol{s}\in\boldsymbol{\mathcal{S}},h\in\mathcal{G}} \sum_{\boldsymbol{s}'} \left(\min_{\boldsymbol{a}\in\mathcal{A}_h} \mathbb{P}(\boldsymbol{s}'|\boldsymbol{s},\boldsymbol{a}) - \min_{\boldsymbol{a}\in\bar{\mathcal{A}}_h} \hat{\mathbb{P}}(\boldsymbol{s}'|\boldsymbol{s},\boldsymbol{a})\right) \\
&\leq \max_{\boldsymbol{s}\in\boldsymbol{\mathcal{S}},h\in\mathcal{G}} \sum_{\boldsymbol{s}'} \left(\mathbb{P}(\boldsymbol{s}'|\boldsymbol{s},\boldsymbol{a}^*_{\boldsymbol{s},h}) - \hat{\mathbb{P}}(\boldsymbol{s}'|\boldsymbol{s},\boldsymbol{a}^*_{\boldsymbol{s},h})\right) \\
&\quad (\text{by } \mathbb{P}(\boldsymbol{s}'|\boldsymbol{s},\boldsymbol{a}^*_{\boldsymbol{s},h}) \geq \min_{\boldsymbol{a}\in\mathcal{A}_h} \mathbb{P}(\boldsymbol{s}'|\boldsymbol{s},\boldsymbol{a})) \\
&\leq S \left\|\mathbb{P} - \hat{\mathbb{P}}\right\|_\infty.
\end{aligned}
$$

We also slightly abuse the infinite norm at the last line and define

$$
\left\|\mathbb{P} - \hat{\mathbb{P}}\right\|_\infty = \max_{\boldsymbol{s},\boldsymbol{s}'\in\boldsymbol{\mathcal{S}},h\in\bar{\mathcal{A}}} |\mathbb{P}(\boldsymbol{s}'|\boldsymbol{s},\boldsymbol{a}) - \hat{\mathbb{P}}(\boldsymbol{s}'|\boldsymbol{s},\boldsymbol{a})|.
$$

Through Hoeffding's inequality shown in Lemma 5, with probability exceeding $1 - \delta$, one has

$$
\left\|\mathbb{P} - \hat{\mathbb{P}}\right\|_\infty \leq \sqrt{\frac{S|\bar{\mathcal{A}}| \log \frac{2S|\bar{\mathcal{A}}|}{\delta}}{2K_1}}.
$$

The above equation shows $0 \leq \hat{\beta}^*_P(|g|) \leq \beta^*_P(|g|)$. Combining the above equations, we have

$$
\left|\beta^*_P(|g|) - \hat{\beta}^*_P(|g|)\right| \leq S \max\left(\eta_P, \sqrt{\frac{S|\bar{\mathcal{A}}| \log \frac{2S|\bar{\mathcal{A}}|}{\delta}}{2K_1}}\right). \tag{50}
$$

Similarly,

$$
\beta^*_R(|g|) = \max_{\boldsymbol{s}\in\boldsymbol{\mathcal{S}},h\in\mathcal{G},n\in\mathcal{N}} \max_{\boldsymbol{a}_1,\boldsymbol{a}_2\in\mathcal{A}_h} (R_n(\boldsymbol{s},\boldsymbol{a}_1) - R_n(\boldsymbol{s},\boldsymbol{a}_2)) \leq \eta_R,
$$

where the inequality is by Eq. (7). Comparing the definition of $\beta^*_R(|g|)$ and $\hat{\beta}^*_R(|g|)$, we have $0 \leq \hat{\beta}^*_R(|g|) \leq \beta^*_R(|g|)$. Therefore,

$$
0 \leq \beta^*_R(|g|) - \hat{\beta}^*_R(|g|) \leq \beta^*_R(|g|) \leq \eta_R. \tag{51}
$$

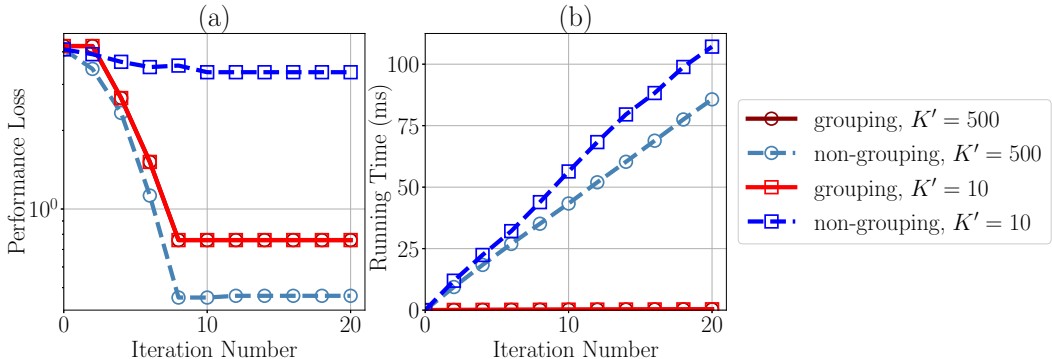

Figure 4: Downlink transmission setting with $S = 5$, $A = 10^5$, and $G = 10$. (a) Performance loss comparison between grouping and non-grouping structure. (b)Running time comparison.

Then we can get Lemma 1 by decreasing and Lipschitz continuity property of the utility function. We have

$$
\begin{aligned}
f^* - \hat{f}^* =& f(|g^*|) - f(|\hat{g}^*|) \quad \text{(by definitions of } f^* \text{ and } \hat{f}^* \text{ in Section 5)} \\
=& f(|g^*|) + (-\hat{f}(|g^*|) + \hat{f}(|g^*|)) + (-\hat{f}(|\hat{g}^*|) + \hat{f}(|\hat{g}^*|)) - f(|\hat{g}^*|) \\
=& (f(|g^*|) - \hat{f}(|g^*|)) + (\hat{f}(|g^*|) - \hat{f}(|\hat{g}^*|)) + (\hat{f}(|\hat{g}^*|)) - f(|\hat{g}^*|)) \\
\leq& (f(|g^*|) - \hat{f}(|g^*|)) + (\hat{f}(|\hat{g}^*|)) - f(|\hat{g}^*|)) \\
& \text{(by } |\hat{g}^*| = \arg\max_{|g|} \hat{f}(|g|) \text{ and } \hat{f}(|g^*|) - \hat{f}(|\hat{g}^*|) \leq 0) \\
\leq& L|\epsilon_{\text{perf}}(|g^*|) - \hat{\epsilon}_{\text{perf}}(|g^*|)| + L|\epsilon_{\text{perf}}(|\hat{g}^*|) - \hat{\epsilon}_{\text{perf}}(|\hat{g}^*|)| \quad \text{(by Eq. (7))} \\
\leq& 2L \max_{|g|} |\epsilon_{\text{perf}}(|g|) - \hat{\epsilon}_{\text{perf}}(|g|)| \\
=& 4L \max_{|g|} \frac{\left| \beta_R^*(|g|) - \hat{\beta}_R^*(|g|) \right|}{1 - \gamma} + 4L \max_{|g|} \frac{\left| \gamma \left( \beta_P^*(|g|) - \hat{\beta}_P^*(|g|) \right) \right|}{(1 - \gamma)^2} \\
& \text{(by definitions of } \epsilon_{\text{perf}}(|g|) \text{ and } \hat{\epsilon}_{\text{perf}}(|g|) \text{ in Eqs. (4) and (49))} \\
\leq& \frac{4L\eta_R}{1 - \gamma} + \frac{4L\gamma S \max\left( \eta_P, \sqrt{\frac{S|\bar{A}| \log \frac{2S|\bar{A}|}{\delta}}{2K_1}} \right)}{(1 - \gamma)^2} \quad \text{(by Eqs. (50) and (51)).}
\end{aligned}
$$

This concludes the proof of Proposition 1. □

## F EMPIRICAL RESULTS

### F.1 LARGER ACTION SPACE

We conducted an experiment under the wireless transmission system with a larger action space. We set $S = 5$, $A = 10^5$, and the number of groups is $G = 10$. The details of the downlink transmission setting are in Appendix B.2. As demonstrated in Fig. 4, when the number of samples is limited, the grouping-based algorithm performs significantly better than the non-grouping method, and the computational complexity is greatly reduced. This simulation implies that the grouping method is applicable to a practical setting with a large action space.

### F.2 GROUPING SELECTION METHOD

We applied the grouping selection method within a downlink transmission scenario, detailed in Appendix B.2. For this purpose, we randomly selected 20 actions from each group, allocating 200 samples for the MDP estimation corresponding to each action. In this specific scenario, addressing

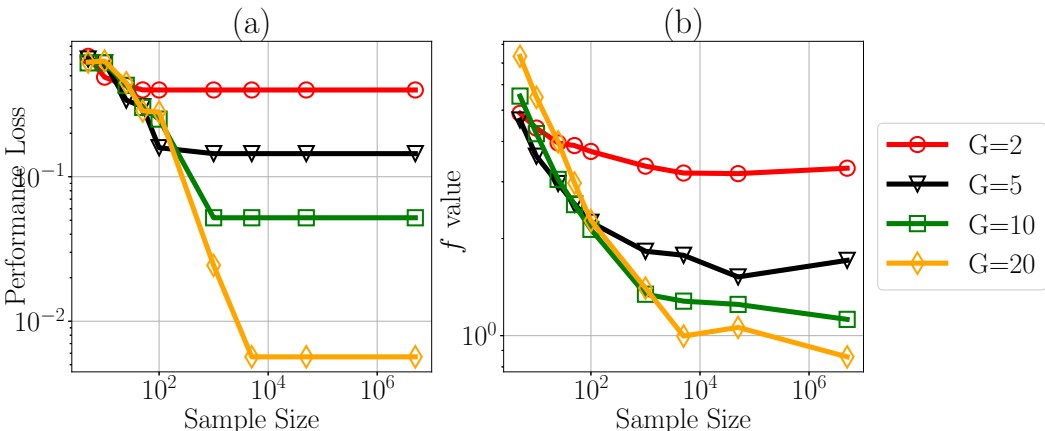

Figure 5: Downlink transmission setting with $A = 1000$, $S = 5$. (a) Performance loss of applying Algorithm 2. (b)Objective function of (P2).

the approximated optimization problem (P2) enables us to effectively identify an optimal grouping function, which balances the trade-off between sample complexity and performance loss. Notably, in conditions where a substantial sample size is available, our optimization approach tends to select a more refined grouping function. Conversely, under sample size constraints, the method inclines towards a coarser grouping approach. This adaptability reflects the capability of the proposed grouping function optimization method in practical settings.

