# OpenReview forum: "Achieving Sample and Computational Efficient Reinforcement Learning by Action Space Reduction via Grouping"
_ICLR.cc/2024/Conference — ICLR 2024 poster_

### Official Review · Reviewer_ND6P · 2023-10-29

**Soundness:** 3 good
**Presentation:** 3 good
**Contribution:** 3 good
**Rating:** 6
**Confidence:** 3

**Summary:**

This paper addressed the issue of explosive state-action spaces in high-dimensional RL problems. More specifically, it proposed to partition the action spaces into multiple groups based on the similarity in the transition distribution and reward function to reduce the size of the action space. Theoretical analysis shows that a finer representation reduces the approximation error resulting from grouping and also introduces higher estimate error when estimating the model. In addition, this work formulated a general optimization problem to select the grouping function.

**Strengths:**

- The proposed grouping strategy for high-dimensional RL problems is novel.

- This paper provides theoretical analysis of the proposed method and the conclusions are interesting.

- The writing is clear and easy to follow.

**Weaknesses:**

- Since this work focuses on MDPs with large action spaces that exhibit group-wise similarity, so it's application could be narrow.

- The current experiments are only conducted on toy tasks. So it's not unclear how would the proposed method perform in more complex tasks.

- Typos: "of each sate-group pair" ==>  state,

**Questions:**

- How would the proposed perform when compared with other kinds of action abstraction methods?

- I am not sure when do we need the proposed method. In which RL tasks do we specific need the proposed method to perform well?

---

> ### Author Response · Authors · 2023-11-20
>
> > Since this work focuses on MDPs with large action spaces that exhibit group-wise similarity, so its application could be narrow. I am not sure when we need the proposed method. In which RL tasks do we specific need the proposed method to perform well?
>
> We respectfully disagree with the statement that the MDPs that exhibit group-wise similarity are narrow.  Our action group method is relevant in numerous practical settings. For instance, actions that correspond to proximate directions can be aggregated into a group in autonomous driving. Another example is congestion control in communication networks, where the actions that send the same (or similar) number of packets are placed in the same group. Further, note that for applications in which grouping functions are not explicitly given by domain knowledge, we can also employ supervised learning to learn the action groups.  For example, we can use more easily accessible offline datasets to classify actions into multiple groups based on the similarity of transition kernels and rewards.
>
>
>
> > The current experiments are only conducted on toy tasks. So it's not unclear how would the proposed method perform in more complex tasks.
>
> Thank you. We have now added an experiment under the downlink transmission scenario in Appendix F.1 to show the applicability of the grouping method when the action space is large ($A=10^5$). These simulations also illustrate that in scenarios where sampling and computational resources are abundant, a non-grouping framework might offer superior performance. However, in situations where such resources are constrained, our grouping framework demonstrates better efficiency and effectiveness.
>
> > Typos: "of each sate-group pair" ==> state.
>
> Thanks for your comment. We have corrected this typo.
>
> > How would the proposed method perform when compared with other kinds of action abstraction methods?
>
> We compare our methods with other action abstractions in the following. One well-researched action abstraction type is temporally related actions from an initial state and terminal state [1].  However, these are specifically targeting the temporally extended action trajectories instead of actions, therefore they are significantly different from our works.
>
> Another line of action abstractions is about factored action space. [2] represents the action space as the binary sequence based on prior knowledge, and learns the value function related to each bit. [3] further uses deep RL to learn the compositional structure of discrete action space to learn better policies and value functions. There are also works that learn the action representations based on prior knowledge that is extracted from expert demonstrations [4]. Without using prior knowledge, [5] learns the underlying structure of actions. However, our work diverges from these methods on applications where domain knowledge of the RL task is available but not yet fully leveraged to effectively balance the complexity and performance loss. Our research targets these applications, demonstrating that an appropriately designed grouping function can effectively reduce complexity with minimal impact on performance.
>
> [1] Riemer, M., Liu, M. and Tesauro, G., 2018. Learning abstract options. Advances in neural information processing systems, 31.
>
> [2] Pazis, J. and Parr, R. Generalized value functions for large action sets. In Proceedings of the 28th International Conference on Machine Learning (ICML-11), pp. 1185– 1192, 2011.
>
> [3] Sharma, S., Suresh, A., Ramesh, R. and Ravindran, B., 2017. Learning to factor policies and action-value functions: Factored action space representations for deep reinforcement learning. arXiv preprint arXiv:1705.07269.
>
> [4] Tennenholtz, G. and Mannor, S., 2019, May. The natural language of actions. In International Conference on Machine Learning (pp. 6196-6205). PMLR.
>
> [5] Chandak, Y., Theocharous, G., Kostas, J., Jordan, S. and Thomas, P., 2019, May. Learning action representations for reinforcement learning. In International conference on machine learning (pp. 941-950). PMLR.

---

> > ### Comment · Reviewer_ND6P · 2023-11-22
> >
> > Thanks for the reply. The responses addressed most of my concerns and I will keep my rating.

---

### Official Review · Reviewer_4oMe · 2023-11-02

**Soundness:** 2 fair
**Presentation:** 4 excellent
**Contribution:** 3 good
**Rating:** 6
**Confidence:** 3

**Summary:**

In large action space tasks, action grouping can make learning tractable. This paper formulates and objective to combine the performance loss due to grouping along with the computational and sample-efficiency benefits of grouping. They propose a practical approximation to this objective, that can provide a strategy for selecting near-optimal yet sample-efficient grouping strategies, altogether.

**Strengths:**

- The paper has a well-written flow and provides all the necessary prerequisities to understand the key contribution and distinction from the prior work.
- The novel insight of cumulatively considering the performance loss due to grouping with limited samples and compute is an important relationship to understand and subsequently help with selecting a strategy for action grouping.
- The practical algorithm makes the complexity of optimal grouping search to be proportional to the number of groups, which makes the problem tractable to solve.

**Weaknesses:**

- There is no empirical demonstration of the practical viability of the practical method to estimate the performance-complexity trade-off. Since this is proposed as a "practical" approximation to the original joint optimization objective, it should be shown how this enables one to select a better action grouping strategy in an RL environment.
- Extension to practical domains: How do the insights and theorems built on tabular value and policy iteration apply to deep RL? This is an important concern because, at the scale of large action spaces, such as those involved in recommender systems, deep RL is the only viable solution.

**Questions:**

- If doing the action abstraction online with RL, how important is the concern about the sample and computational complexity of estimating the abstractions? Wouldn't the sample efficiency be subsumed within the sample complexity of RL itself?
- If the groupings are learned, how can the lower-level policy be feasibly obtained using domain knowledge? With the group compositions potentially changing non-stationarily, the lower level policy might itself be very challenging to learn or provide with domain knowledge.

---

> ### Author Response · Authors · 2023-11-20
>
> > There is no empirical demonstration of the practical viability of the practical method to estimate the performance-complexity trade-off. Since this is proposed as a "practical" approximation to the original joint optimization objective, it should be shown how this enables one to select a better action grouping strategy in an RL environment.
>
> Thanks for your suggestion on the empirical demonstration of the proposed practical method. To address this, we applied the grouping selection method in the downlink transmission setting (See figure in Appendix F. 2). In this particular setting, solving the approximated optimization problem (P2) allows us to identify the most effective grouping function to balance the tradeoff between the sample complexity and performance loss. For instance, with a large sample size, the optimization process tends to select a more refined grouping function ($G=20$). On the other hand, in scenarios where the sample size is constrained, the method gives a coaster grouping function.
>
> > Extension to practical domains: How do the insights and theorems built on tabular value and policy iteration apply to deep RL? This is an important concern because, at the scale of large action spaces, such as those involved in recommender systems, deep RL is the only viable solution.
>
> There are scenarios where deep RL is not an appropriate tool for solving problems. Deep RL uses samples to simultaneously learn the relationships between state-action pairs and a good policy. However, when the domain knowledge of the RL task is given, how to use domain knowledge to reduce the sample complexity is not known in deep RL. We target these applications and show that the grouping function can effectively reduce the complexity while inducing little performance loss.
>
> Our methodology is quite general. So, we could use domain knowledge and apply action grouping as the preprocessing step to reduce the dimensionality of the action space, thus making deep neural network training more feasible and efficient. Thank you for the suggestion, this is a good problem to investigate further as a future direction.
>
> > If doing the action abstraction online with RL, how important is the concern about the sample and computational complexity of estimating the abstractions? Wouldn't the sample efficiency be subsumed within the sample complexity of RL itself?
>
> We agree that the sample efficiency could be dominated by RL algorithms instead of estimating the abstractions. But note that our solution is indeed focused on reducing the complexity of the RL itself. Our framework is built upon the assumption that action grouping functions are predetermined based on specific domain knowledge. This allows us to concentrate on the efficiency and performance of the RL algorithms, rather than the complexities involved in estimating these abstractions. In Theorem 1, we investigated the performance degradation caused by limited samples as well as action grouping. Also, the proposed grouping-selection method in Section 5 also aims to balance the performance degradation and computational/sample complexity.
>
>
> > If the groupings are learned, how can the lower-level policy be feasibly obtained using domain knowledge? With the group compositions potentially changing non-stationary, the lower level policy might itself be very challenging to learn or provide with domain knowledge.
>
> Note that from Theorem 1, we show that when actions in the same group are similar, the approximation error is small. Hence, when the transition kernel and rewards within a group are closely aligned, variations in lower-level policies do not significantly impact overall performance.
>
> Our current focus on utilizing a domain knowledge-based grouping function is an initial step in our research. The exploration of learning the lower-level policy, particularly in dynamic MDP environments, can be an exciting direction for future work.

---

### Official Review · Reviewer_oqpX · 2023-11-03

**Soundness:** 2 fair
**Presentation:** 2 fair
**Contribution:** 2 fair
**Rating:** 5
**Confidence:** 3

**Summary:**

This paper studies abstraction in reinforcement learning at the level of the action space. The authors propose to group similar action under an action grouping method, using this grouping method they can significantly reduce the size of the action space and lead to faster convergence.
The authors also analyze the loss induced by this grouping mechanism and show that it depends on the approximation error of the grouping method and the estimation error of the MDP induce by the grouping method.
It is then shown that it is possible to strike a balance between these two terms and that using a abstraction even if not perfect can lead to better performance.
Finally the author show how to optimize the grouping function to minimize this error.

**Strengths:**

This paper studies action abstraction, historically it has been less studies than state abstraction or abstractions over multiple timesteps.
Overall the paper is well written and the main ideas are explained in detail.
The main result of the Theorem 1 is also interesting and new as far as I known.

**Weaknesses:**

The notation used in the paper were often not precise and made it more difficult to understand the work presented:
- In section 3.1 it is not explained if the state space and action are discrete, finite or bounded.
Because of that is also unclear if |g| defined in section 3.2 can be infinite. In the same section for the definition of $\pi^i$ should it be $\{h\}$ instead of $h$?
- It would also be helpful to properly define the input of $\mathbb{P}_1$ and $\mathbb{P}_2$ to highlight that one applies on the original action space and the other applies on the lifted action space.
- How is K defined in section 3.3? I don't understand the notation used.

My big issue with this paper is that I do not understand how Algorithm 1 can lead to an almost optimal policy. It seems like an exploration term is missing to ensure a good coverage of the action / state space. Right now the algorithm could be stuck in a local minima while being able to accurately model the lifted MDP. Please let me know if I missed something

"However, the relationship between the performance loss and the goruping function is surprising. A grouping function with a larger number of groups can sometimes achieve better performance."
I don't think this is particularly surprising when this is the reason why we use abstractions! Similar results exisit in the context of state abstraction, when looking at finite time guarantees while abstraction introduces an approximation error it can also lead to much better performance. [1, 2]

Section 5.2, given that $\mathbb{P}$ and R are bounded isn't assumption 4 always verified?

[1] A unifying framework for computational reinforcement learning theory, Lihong Li
[2] Approximate Exploration through State Abstraction, Taiga et al. arxiv 2018

**Questions:**

Can you explain to me how Algorithm 1 is working without doing exploration?

---

> ### Author Response · Authors · 2023-11-20
>
> > In section 3.1 it is not explained if the state space and action are discrete, finite or bounded. Because of that is also unclear if |g| defined in section 3.2 can be infinite.
>
> Thanks for pointing out the need for clarity in Section 3.1. We confirm that in our current framework, both the state and action spaces are assumed to be discrete and finite. This assumption aligns with our grouped model, where $|g|$, the size of each action group, is also considered as a discrete and finite function. We have revised the paper accordingly.
>
> We also acknowledge the potential of extending our model to continuous action spaces. In such scenarios, we propose to utilize domain knowledge to segment the continuous action space into multiple discrete and finite groups. This adaptation would allow us to apply our current algorithms to the lifted MDP, effectively reducing the sample complexity inherent in continuous action space.
>
> > It would also be helpful to properly define the input of $\mathbb{P}_1$ and $\mathbb{P}_2$ to highlight that one applies to the original action space and the other applies to the lifted action space.
>
> Thank you for your suggestions on definitions of $\mathbb{P}_1$ and $\mathbb{P}_2$. $\mathbb{P}_1:\mathcal{S}\times \mathcal{G}\rightarrow \mathcal{S}$ is the transition probability from the state and group pair belonging to the state space and the lifted actions space to the next state. $\mathbb{P}_2:\mathcal{S}\times \mathcal{A}\rightarrow \mathcal{S}$ is the transition probability from the state and action pair belonging to the state space and the primitive actions space to the next state. We have revised the paper accordingly.
>
>
> > How is K defined in section 3.3? I don't understand the notation used.
>
> $K$ is the number of samples to evaluate the transition kernel of all state-group pairs. Essentially, $K$ can be regarded as the total required samples for estimating the lifted MDP.
>
> > My big issue with this paper is that I do not understand how Algorithm 1 can lead to an almost optimal policy. It seems like an exploration term is missing to ensure good coverage of the action/state space. Right now the algorithm could be stuck in a local minima while being able to model the lifted MDP accurately. Please let me know if I missed something.
>
> Note that once we have modeled the lifted MDP accurately, we do not need to use exploration. We can simply use value iteration to converge to the nearly optimal solution (which is exactly what Algorithm 1 does). Our approach of grouping provides a tradeoff between sub-optimality and complexity. The sub-optimality occurs because of (i) grouping, (ii) the estimation error of the lifted MDP, and (iii) the algorithmic error. For the approximation error caused by grouping,  we show that when actions in the same group have similar transition probability and rewards (i.e., $\beta_P$ and $\beta_R$ are small), the performance loss due to different lower-level policies within groups is small. The estimation error of the lifted MDP is only proportional to the group size and inversely proportional to the number of samples. The algorithmic error decreases exponentially w.r.t. the iteration number.

---

> > ### Author Response · Authors · 2023-11-20
> >
> > > "However, the relationship between the performance loss and the grouping function is surprising. A grouping function with a larger number of groups can sometimes achieve better performance." I don't think this is particularly surprising when this is the reason why we use abstractions! Similar results exist in the context of state abstraction when looking at finite time guarantees while abstraction introduces an approximation error it can also lead to much better performance. [1, 2]
> >
> > Thanks for your comments and for providing these references. The “surprising” part that we were referring to was not that fewer groups would lead to lower complexity at the cost of performance. The surprising part was that in scenarios with a limited number of samples, a smaller number of groups may in fact prove to be more effective than using a larger number of groups. This is because even though the approximation error is smaller in the case with a large number of groups, the estimation error caused by the limited samples for lifted MDP estimation with a large number of groups dominates the approximation error. Therefore, the
> > overall performance loss of the smaller number of groups case is still significantly larger than the setting with a large number of groups. To the best of our understanding this has not been shown in [1] and [2]. Moreover, we want to emphasize that since we quantitatively analyzed the sampling/computational complexity to learn a good policy with action grouping, we can additionally explore a computationally efficient approach to find the nearly optimal grouping function that carefully balances the complexity and performance loss.
> >
> >
> > > Section 5.2, given that $\mathbb{P}$ and R are bounded isn't assumption 4 always verified?
> >
> > Yes, we agree that Assumption 4 always holds. We will appropriately rewrite $\eta_P=\max_{s,h,g(a_1)=g(a_2)}\|\mathbb{P}(\cdot|s,a_1)-\mathbb{P}(\cdot|s,a_2)\|$ and $\eta_R=\max_{s,h,g(a_1)=g(a_2)}R(s,a_1)-R(s,a_2)$.
> >
> > > Can you explain to me how Algorithm 1 is working without doing exploration?
> >
> > Algorithm 1 contains two phases: (i) the estimation of the lifted MDP, and (ii) planning based on the estimated MDP. Note that once the estimation of the lifted MDP is accomplished, there is no further need to explore the state-action space extensively. Instead, we can apply planning algorithms to find the nearly optimal policy. As indicated by Theorem 1, when the actions within the same group exhibit significant similarity, the performance loss caused by grouping (defined as approximation error) is small, thereby ensuring that the output policy is the nearly optimal policy.

---

### Official Review · Reviewer_62eA · 2023-11-10

**Soundness:** 3 good
**Presentation:** 3 good
**Contribution:** 2 fair
**Rating:** 6
**Confidence:** 3

**Summary:**

The paper proposes an approach to achieve better sample efficiency in high-dimensional RL. To that end, the authors find a low-rank representation of the high-dimensional MDP by partitioning the action space in groups according to the predictive capabilities of the transition and reward models. In particular, the presented approach reduces complexity (induced by the abstraction) while addressing the decline in performance due to abstraction errors.  For that, the authors build a linear decomposition model to capture these relationships and optimize over the grouping strategy to strike this trade-off.  The authors also establish (tight) bounds for the performance drop due to action grouping. Finally, the paper presents a general optimization problem that considers the trade-off between performance loss and complexity.

**Strengths:**

The paper proposes an interesting idea (group the action space according to the environment model) and addresses a challenging problem (high dimensional RL). The paper is also generally well written and states the details of the derivations clearly. The authors review the existing work in detail. The inclusion of theoretical results solidifies the claims made in the paper.

**Weaknesses:**

The paper can be improved (and the reviewer is happy to raise the score if these points are met) if the authors would provide some empirical results. It is yet not apparent if the ideas of the presented approach ramifies in practical advances. As applications, the authors already mention control systems with potentially millions of actions available at each step or recommender systems with large number of potential items. Another possibility would be to test on muscular systems [1], since action space grouping would be quite effective for tasks involving such systems.

In general, it would also help to understand the ramifications of the assumptions of this paper for potential applications. Can this method solve all possible RL tasks, or is a set of tasks excluded? In the paper, the authors write: “We specifically focus on MDPs with large action spaces that exhibit group-wise similarity, where only approximate grouping strategies of the action space are available”. Which particular tasks are meant in this sentence?

[1] [https://github.com/MyoHub/myosuite](https://github.com/MyoHub/myosuite)

**Questions:**

See weakness section.

---

> ### Author Response · Authors · 2023-11-20
>
> > The paper can be improved (and the reviewer is happy to raise the score if these points are met) if the authors would provide some empirical results. It is yet not apparent if the ideas of the presented approach ramifies in practical advances. As applications, the authors already mention control systems with potentially millions of actions available at each step or recommender systems with large number of potential items. Another possibility would be to test on muscular systems [1], since action space grouping would be quite effective for tasks involving such systems.
>
> We conducted an experiment under the wireless transmission system with a larger action space. We set $S=5$, $A=10^5$, and the number of groups is $G=10$. As shown in the figure in Appendix F.1, when the number of samples is limited, the grouping-based algorithm performs significantly better than the non-grouping method, while the computational complexity is greatly reduced. This simulation implies that the grouping method is applicable to a practical setting with a large action space.
>
> Our approach lends itself to many practical applications, especially in scenarios where the groups can be constructed such that the number of groups is much smaller than the number of actions. The focus of this paper is to solve the problem given a set of groups. However, the framework can be extended to the case when the groups themselves need to be determined by leveraging domain knowledge and empirical works in this area. In this case, we can construct an initial rough grouping function based on domain knowledge, and iteratively update the lower-level policy and grouping function to learn a refined grouping function under acceptable complexity. This is an interesting direction in the future.
>
> > In general, it would also help to understand the ramifications of the assumptions of this paper for potential applications. Can this method solve all possible RL tasks, or is a set of tasks excluded? In the paper, the authors write: “We specifically focus on MDPs with large action spaces that exhibit group-wise similarity, where only approximate grouping strategies of the action space are available”. Which particular tasks are meant in this sentence?
>
> Assumption 1 requires the rewards to be bounded, which is a commonly used assumption and is satisfied in a broad set of practical settings. Assumption 2 requires the utility function to be monotonically decreasing with all variables. These assumptions are reasonable because we prefer the grouping function that causes lower performance loss and lower sample/computational complexity. Assumption 3 requires the utility function to be Lipschitz continuous with respect to the first variable, that is the utility function changes slightly when the performance loss changes.
>
> We can apply this work to applications as long as we can divide actions into multiple groups where actions within the same group exhibit similar transition probabilities and rewards, as indicated by small values of  $\beta_P$ and $\beta_R$ in Eqs. (1) and (2). To illustrate, consider the congestion control of wireless access systems [1]. Here all actions that send the same number of packets are in the same group. Additionally, in downlink transmission scenarios where the base station sends packets to multiple users, actions can be grouped based on users' characteristics.  In such settings, actions that involve sending packets to users with similar profiles can be considered as parts of the same group.
>
> We admit there are some tasks for which the action grouping is not applicable. For example, when all actions are distinctly different, it is impossible to find a valid grouping function that can reduce the “effective” action space without inducing significant performance loss.
>
> [1] Ghaffari, Ali. "Congestion control mechanisms in wireless sensor networks: A survey." Journal of network and computer applications 52 (2015): 101-115.

---

> > ### Comment · Reviewer_62eA · 2023-11-22
> >
> > Thank you for the clarifications. I agree that discovering the grouping function would be an enormous extension of this work. I raised my score.

---

### Meta-Review · Area_Chair_K85x · 2023-12-08

**Metareview:**

The research addresses challenges in high-dimensional reinforcement learning by proposing an action grouping strategy for managing explosive state-action spaces. The approach reduces the action space size by partitioning actions based on transition distribution and reward function similarities. The theoretical analysis emphasizes a delicate trade-off: finer grouping diminishes approximation errors but introduces higher estimation errors. The authors formulate an optimization problem to balance these factors, and a practical approximation is presented for selecting near-optimal, sample-efficient grouping strategies. The goal is to enhance tractability and performance in large action spaces.

The paper studies an innovative approach to high-dimensional reinforcement learning, introducing a novel grouping strategy. Its clear and accessible writing style ensures a thorough understanding of key concepts and detailed derivations. The robust theoretical analysis provides valuable insights, notably considering cumulative performance loss due to grouping with limited resources. The practicality of the research is highlighted through the introduction of an algorithm that enhances tractability, making the proposed strategy applicable in real-world scenarios.

On the other hand, the paper lacks empirical demonstrations of the proposed method's practical effectiveness, raising concerns about its real-world applicability. Furthermore, focusing on MDPs with large action spaces exhibiting group-wise similarity may limit its relevance. The limited experimental scope, restricted to toy tasks, adds uncertainty about the method's performance in more complex scenarios. Addressing these issues through empirical evidence and broader experimentation would enhance the paper's credibility.

The authors did a good job of addressing the reviewers' concerns.
We propose accepting the paper for publication, with the understanding that the authors will address the identified weaknesses and incorporate any necessary clarifications or improvements in the final revision.

**Justification For Why Not Higher Score:**

The paper has some limitations, including the lack of empirical evidence, narrow focus on specific types of MDPs, and limited experimental scope on toy tasks.

**Justification For Why Not Lower Score:**

Despite identified weaknesses, the paper's innovative approach to high-dimensional reinforcement learning with a novel grouping strategy and theoretical insights, alongside potential real-world applications, provides a foundation for consideration with suggested improvements through additional experiments.

---

### Decision · Program_Chairs · 2024-01-16

Accept (poster)